# Reduced-Rank Outcome Compression for Causal Policy Optimization

**Ezinne Nwankwo**                                          *ezinne_nwankwo@berkeley.edu*
*Department of Electrical Engineering and Computer Science*
*University of California, Berkeley*

**Michael I. Jordan**                                          *jordan@cs.berkeley.edu*
*Department of Electrical Engineering and Computer Science*
*University of California, Berkeley*

**Angela Zhou**                                          *zhoua@usc.edu*
*Department of Data Sciences and Operations*
*University of Southern California*

**Reviewed on OpenReview:** *https://openreview.net/forum?id=WQhOaY4yPC*

## Abstract

Evaluating the causal impacts of possible interventions is crucial for informing decision-making, especially towards improving access to opportunity. If causal effects are heterogeneous and predictable from covariates, then personalized treatment decisions can improve individual outcomes and contribute to both efficiency and equity. In practice, however, causal researchers do not have a single outcome in mind a priori and often collect multiple outcomes of interest that are noisy estimates of the true target of interest. For example, in government-assisted social benefit programs, policymakers collect many outcomes to understand the multidimensional nature of poverty. The ultimate goal is to learn an optimal treatment policy that in some sense maximizes multiple outcomes simultaneously. To address such issues, we present a data-driven dimensionality-reduction methodology for multiple outcomes in the context of optimal policy learning with multiple objectives. We learn a low-dimensional representation of the true outcome from the observed outcomes using reduced rank regression. We develop a suite of estimates that use the model to denoise observed outcomes, including commonly-used index weightings. These methods improve estimation error in policy evaluation and optimization, including on a case study of real-world cash transfer and social intervention data. Reducing the variance of noisy social outcomes can improve the performance of algorithmic allocations.

## 1 Introduction

Causal inference aims to describe the impact of treatments both on aggregate outcomes and on the trajectories of individuals. When treatment effects are heterogeneous, varying systematically across individuals with different covariates, targeting treatment toward those most likely to benefit can improve overall welfare. The general problem of *policy learning* is to identify optimal treatment policies using observed data collected under varying treatment assignments (Athey & Wager, 2020).

Policy learning is especially consequential when the units of analysis are people. In many high-stakes applications such as commercial recommendation systems, healthcare and welfare benefits programs, algorithmic decisions can have real impacts but the social outcome data that feeds these algorithms are difficult to measure and predict. Complex social outcomes such as poverty, patient health and user satisfaction are all multidimensional constructs that researchers approximate through noisy proxy measurements and they introduce three fundamental data challenges. First, complex social phenomena are difficult to define precisely, leading researchers to measure many different outcomes. Second, variability among humans makes these

outcomes hard to predict. Third, having many outcomes can reduce actionability by requiring decision makers to specify or choose scalarization weights.

For example, the Fragile Families Challenge—a high-profile "common task framework" study—evaluated whether machine learning could accurately predict important life outcomes (Salganik et al., 2020b). Despite hundreds of teams applying state-of-the-art models to a rich dataset, predictions were only modestly accurate and did not outperform a simple baseline model that used four expert-selected features. Beyond the caution this result raises for predicting social outcomes, it underscores that human-related data often involve noisy measurements, ill-defined latent constructs, and multiple outcomes.

This paper focuses on the challenge of multiple noisy outcomes and using them for precise estimation and learning, a problem that arises across many domains. In e-commerce pricing, search, and recommendation systems, learning algorithms increasingly rely not only on click streams but also on various forms of "implicit feedback" that capture users' purchasing propensity (Garrard et al., 2023; Yao et al., 2022; Tripuraneni et al., 2023). In healthcare, clinicians consider multiple outcomes, including clinical improvement and potential side effects—when tailoring treatments to individual patients (Bastani, 2021; Chen et al., 2021). In social benefits programs, our motivating example, poverty is a dynamic and multidimensional issue (Anand & Sen, 1997). When allocating resources, measured outcomes often capture only subsets of the underlying construct. Economic variables, social and emotional factors, and other dimensions all contribute to the experience of poverty (Bossuroy et al., 2022b; Banerjee et al., 2022; Blattman et al., 2015).

Our study of this issue is motivated by the challenges of prediction in social settings (Salganik et al., 2020a; Lundberg et al., 2024; Liu et al., 2025). To give an example, data from the Sahel poverty graduation program illustrates these same challenges, which occur more broadly (Bossuroy et al., 2022a). (We revisit this data in our later experiments section). Researchers conducted a randomized control trial to provide monetary and psychosocial support for households experiencing poverty. Over 100 survey outcomes are collected and combined using a weighted average to form index outcomes that reduce dimensionality and noise. For example, the daily consumption index is constructed as a weighted average between different consumption metrics such as weekly home food consumption, weekly expenditure on food away from home, health and education expenditure, and yearly celebration expenditure. A preliminary analysis where we regress this index on the full set of baseline covariates, collected one year prior, shows that these combined outcomes exhibit high variance with an $R^2$ value of 0.10. On other indices in the data we find that when controlling for the full set of available baseline covariates, we observe a median $R^2$ value of 0.10, with no index exceeding 0.26. While not surprising in isolation, this persistent residual variance has direct consequences for policy learning: noisy outcome measurements inflate the variance of value estimates, making it harder to identify which treatment rules genuinely improve welfare. The averaging or index-construction methods used in the field often ignore learnable structure of the measured variables and correlations across outcomes.

Prior work has explored dimensionality reduction techniques (McKenzie, 2005), highlighted specific challenges posed by multiple outcomes (Ludwig et al., 2017; Björkegren et al., 2022), and multiobjective policy learning (Boominathan et al., 2020), but exploiting the low-rank structure of outcomes for denoising has not been explored in the policy learning literature.[1] Relative to prior work on low-rank outcome models (Agarwal et al., 2020) based on model-based estimation alone, we explore rank reduction for causal estimators that incorporate inverse propensity reweighting. Relative to standard variance reduction from double-robustness or shrinkage-based approaches that clip propensity weights to trade-off bias and variance (Su et al., 2020; Wang et al., 2017), we explore further denoising of factual outcomes to reduce variance from irreducible outcome noise. Our work is the first to integrate reduced-rank outcome denoising directly into the policy learning pipeline, yielding estimators that can handle noisy, high-dimensional outcome settings common in real-world policy evaluation.

We address this gap with three contributions. First, we propose a latent-variable framework that denoises high-dimensional outcomes by recovering low-rank structure via reduced-rank regression. We refer to this as outcome compression: by exploiting shared structure across multiple noisy outcomes, we obtain denoised

---

[1]Alternative factor models that have been considered, such as probabilistic PCA (Tipping & Bishop, 1999) and penalized factor models (Yuan et al., 2007), are either poorly suited to heterogeneous covariate-adjusted outcomes or computationally burdensome and prone to rank overestimation. We discuss these in detail in Appendix A.1.

quantities that reduce measurement variance and provide more stable targets for policy learning. Second, we introduce a regression-based control-variate estimator that reduces variance and is compatible with denoising. Third, through simulated and real policy-learning tasks, we demonstrate consistent gains in policy evaluation accuracy and downstream policy quality. Our empirical results and robustness checks illustrate the benefits of our shrinkage approach that trades off potential model bias from the variance of multiple outcomes — which is nontrivial in real-world, noisy-outcome settings. Our CV family includes a raw-outcome version closely related to doubly robust estimation and a denoised shrinkage version that can improve finite-sample performance when the reduced-rank outcome model is accurate, but we find that no single denoised estimator dominates uniformly. Given the ubiquity of multiple, noisy outcomes in practice, our work demonstrates that leveraging shared structure in the outcomes via low-rank assumptions can reduce variance and improve policy evaluation and learning.

## 2 Problem Setup and Related Work

We begin by summarizing our problem setup. Our data (randomized or observational) consists of tuples of random variables, $\{(X_i, T_i, Y_i) : i = 1, \ldots, n\}$, with $p$-dimensional covariates $X_i \in \mathbb{R}^p$, a binary treatment assignment $T_i \in \{0, 1\}$, and vector-valued outcomes $Y_i \in \mathbb{R}^k$. (The model is easily generalized to multiple treatments, but we start with binary treatments for brevity). Let $Y_i(0)$ and $Y_i(1)$ denote the Neyman-Rubin potential outcomes for unit $i$. We assume consistency, $Y_i = Y_i(T_i)$, so that the observed outcome equals the potential outcome corresponding to the treatment actually received. We also assume ignorability, $Y(T) \perp T \mid X$, meaning that treatment assignment is fully explained by observed covariates, and the stable unit treatment value assumption (SUTVA), that treatments of others do not affect one individual's outcomes. These assumptions are standard, but not innocuous, in causal inference. Ignorability holds by design, for example, in randomized controlled trials.

We define the outcome model $\mu_t(X) = \mathbb{E}[Y(t) \mid X]$, and the propensity score $e_t(X) = P(T = t \mid X)$. A policy $\pi(t \mid X) \in [0, 1]$ describes the probability of assigning treatment $t$ to an individual with covariates $X$. Our goal is both to evaluate alternative treatment assignment rules and optimize for the best one. We restrict the policy $\pi$ to a chosen function class, such as a parametrized probabilistic classifier, to ensure interpretability and generalization. The *policy value estimand* is the population expectation of potential outcomes, under treatment assignment induced by $\pi$. We consider two settings for multi-objective policy learning:

1. Scalarization of the observed outcomes $Y(t) \in \mathbb{R}^{n \times k}$

2. Scalarization of estimated latent factors alone $Z(t) \in \mathbb{R}^{n \times r}$, when $Y(t)$ is high-dimensional and $Y(t) = g(Z(t))$ for some mapping $g$.

Given a weighting vector $\rho \in \mathbb{R}^k$ for observed outcomes or $\rho \in \mathbb{R}^r$ for latent factors, the corresponding scalarized policy values are:

$$V_Y(\pi) = \mathbb{E}[\rho^\top Y(\pi)] \text{ or } V_Z(\pi) = \mathbb{E}[\rho^\top Z(\pi)], \tag{1}$$

where $\mathbb{E}[\rho^\top Y(\pi)] = \sum_t \mathbb{E}[\pi(t \mid X)\rho^\top \mathbb{E}[Y(t) \mid X]]$ and analogously for $\mathbb{E}[\rho^\top Z(\pi)]$. Unless otherwise noted, we adopt the convention that the outcomes $Y_i$ represent losses, so lower values correspond to better outcomes.

### 2.1 Estimation in causal inference

We briefly review relevant causal inference estimators for off-policy evaluation and optimization in the case of a *single outcome*. Standard approaches include the direct method, inverse propensity weighting, and doubly robust estimators, each of which adjusts in different ways for counterfactual outcomes. One can also interpret the below estimators as vector-valued, i.e. applying each standard strategy to each outcome separately.

**Direct Method (DM).** The direct method estimates the relationship between the potential outcomes $Y(t)$ conditional on the covariates $X$ and the treatment $T$. In particular, we use outcome regression to learn $\hat{\mu}_t(X) = \mathbb{E}[Y(t)|X]$ for $t = 0, 1$.

A standard estimator based on the above regression models is:

$$\hat{V}_Y^{DM}(\pi) = \sum_t \mathbb{E}_n\left[\pi(t \mid X)\hat{\mu}_t(X)\right], \tag{2}$$

where we use a logistic parameterization for the policy, such as $\pi(t \mid X) = sigmoid(\theta^\top X)$.

**Inverse Propensity Weighting (IPW).** If the regression model $\hat{\mu}_t(X)$ is misspecified relative to the true outcome model, then the direct method can be biased. Inverse propensity weighting instead reweights the observed outcomes by the inverse of the propensity scores $e_t(X)$, yielding an unbiased estimator. However, the weights $\frac{1}{e_t(X)}$ can inflate variance, especially when propensities are small. We estimate the propensity scores $\hat{e}_t(X)$ by learning a logistic regression model from observed data.

The standard IPW estimator is:

$$\hat{V}_Y^{IPW}(\pi) = \sum_t \mathbb{E}_n\left[\pi(t \mid X)\frac{\mathbb{I}\left[T = t\right]Y}{\hat{e}_t(X)}\right]. \tag{3}$$

**Doubly Robust (DR).** The doubly robust estimator (Bang & Robins, 2005; Scharfstein et al., 1999) combines DM and IPW. It remains unbiased if either the propensity score model or the outcome model is correctly specified.

The standard DR estimator is:

$$\hat{V}_Y^{DR}(\pi) = \sum_t \mathbb{E}_n\left[\pi(t \mid X)\left[\frac{\mathbb{I}\left[T = t\right]\left(Y - \hat{\mu}_t(X)\right)}{\hat{e}_t(X)} + \hat{\mu}_t(X)\right]\right], \tag{4}$$

which achieves this doubly robust property.

## 2.2 Reduced Rank Regression for Potential Outcomes

**Dimensionality-reducing factor modeling**. Reduced rank regression (RRR) provides a natural dimension-reducing model for multivariate outcomes that corresponds naturally to our setting, via the "multiple causes to the multiple indicators" (MIMIC) interpretation (Reinsel & Velu, 1998). The underlying generative model for RRR posits a set of unobserved "latent" variables (interpreted as hypothetical constructs) that generate observed outcomes through linear transformations. The RRR model is as follows:

$$Y_i = HX_i + \epsilon_i, \quad i = 1, \ldots, n,$$

where $Y_i \in \mathbb{R}^k$ is a vector of outcomes, $X_i \in \mathbb{R}^p$ is a vector of covariates, $H \in \mathbb{R}^{k \times p}$ is the coefficients matrix, and $\epsilon_i \in \mathbb{R}^k$ is a vector of the error terms.

A key structural assumption of reduced rank regression is that the coefficient matrix $H$ is of lower rank than the original coefficient matrix.

**Assumption 1** (Low rank regression coefficients). $rank(H) = r \ll min(k, p)$.

The low-rank structure in Assumption 1 is motivated by the domains we study (e.g., social data), where outcomes often cluster into conceptual categories. Udell & Townsend (2019) show that big data matrices are often approximately low rank. Later in our experiments, we include robustness checks and our method still performs well when Assumption 1 is not true: in the presence of multiple outcomes, it provides useful structural regularization nonetheless.

Under Assumption 1, $H$ factors into lower-dimensional matrices:

$$H = AB,$$

where $A \in \mathbb{R}^{k \times r}$ and $B \in \mathbb{R}^{r \times p}$, each with rank $r$. The $r$ columns of $A$ span the column space of $H$ and the $r$ rows of $B$ span the row space of $H$. Thus the model becomes

$$Y_i = A(BX_i) + \epsilon_i, \quad i = 1, \ldots, n.$$

This can be interpreted as a factor model with $Z := BX$. Reinsel & Velu (1998) derive the maximum likelihood estimates $\hat{A}$ and $\hat{B}$ for the reduced rank model:

$$\hat{A} = \Gamma^{-1/2}[\hat{V}_1, \ldots, \hat{V}_r], \quad \hat{B} = [\hat{V}_1, \ldots, \hat{V}_r]^\top \Gamma^{1/2} \hat{\Sigma}_{yx} \hat{\Sigma}_{xx}^{-1}, \tag{5}$$

where $\hat{\Sigma}_{yx} = \frac{1}{n} Y X^\top$, $\hat{\Sigma}_{xx} = \frac{1}{n} X X^\top$, and $V_j$ is the eigenvector corresponding to the j-th largest eigenvalue $\hat{\lambda}_j^2$ of $\Gamma^{1/2} \hat{\Sigma}_{yx} \Sigma_{xx}^{-1} \hat{\Sigma}_{xy} \Gamma^{-1/2}$ and we choose $\Gamma = \hat{\Sigma}_{\epsilon\epsilon}^{-1}$. These estimators are optimal and asymptotically efficient under standard normality assumptions. See Bunea et al. (2011) for a discussion of the relationship of this model to the truncated singular value decomposition.

**Reduced rank regression for potential outcomes**.

To apply this model in our causal setting, we posit separate RRR models for treatment and control groups, reflecting that observed outcomes are noisy realizations of potential outcomes.

**Assumption 2** (Multivariate normal $X$). *For $t = 1, 0$, we assume that the covariates are $X \sim N(M, \Sigma_x)$.*

**Assumption 3** (Potential outcomes satisfy reduced-rank regression assumptions).

$$Z(t) = B_t X + U$$
$$Y(t) = A_t Z(t) + \epsilon = A_t(B_t X) + (\epsilon + A_t U),$$

where $U \sim N(0, \Sigma_U)$ and $\epsilon \sim N(0, \Sigma_\epsilon)$. We estimate the latent and observed potential outcomes via plug-in estimators:

$$\hat{Z}_t := \hat{B}_t X, \tag{6}$$
$$\hat{\mu}_t^{RR}(X) := \hat{A}_t \hat{B}_t X, \tag{7}$$

This structure formalizes the common applied practice of constructing index variables, or simple averages of conceptually related outcomes, and provides a data-driven way to improve upon these heuristics.

## 2.3 Multi-objective evaluation and learning

When outcomes are vector-valued, decision-makers use scalarization to reduce the multi-objective structure and solve policy optimization problems. We formalize this process with the assumption below.

**Assumption 4** (Decision-maker preferences). *We assume that the decision-maker specifies weighting vector $\rho \in \mathbb{R}^k$ over the observed outcomes $Y$.*

Many existing ways of combining multiple outcomes into one measure can be represented as scalarizations of the multiple outcomes. An example would be index variables that appear in randomized controlled trials (RCTs) in development economics. Each noisy measured outcome is assigned to a more general category, such as "financial well-being" or "educational progress". The index simply averages the outcomes within the same category. We can represent this in our model via restrictions on the $\rho$ vector.[2].

In this scalarized setting, the optimal policy solves

$$\pi^* \in \arg\min \mathbb{E}[\rho^\top Y(\pi)].$$

Our methods reduce variance in evaluating and optimizing such policies. This is meant to be an improvement over existing index-based approaches commonly used in economics and multi-objective policy learning. Later, in our experiments, we simulate random values for $\rho$ to model this setting in order to illustrate that our method does not depend on the choice of weighting vector, rather it performs variance reduction on top of the dimension reduction that scalarization provides.

---

[2]Namely, consider $\rho$ vectors where every outcome in an index has the same coefficient. Let $\mathcal{I}$ be the set of indices and $I(j) \colon [k] \mapsto [|\mathcal{I}|]$ describes the index of a variable $Y_j$. Then this common practice corresponds to $\{\rho \in \mathbb{R}^k \colon \rho_j = \rho_{j'} \text{ if } I(j) = I(j'), \ \forall j, j' \in [k]\}$.

Table 1: Estimator family in Section 3. Rows correspond to the estimation strategy; columns indicate whether the estimator uses the raw scalarized outcome $\rho^\top Y$ or the denoised reduced-rank prediction $\rho^\top \hat{\mu}_t^{RR}(X)$.

| Estimator type | Raw outcome signal | Denoised reduced-rank signal | Purpose in paper |
|---|---|---|---|
| DM | $\hat{V}_Y^{DM}(\pi)$ in eq. (2) | $\hat{V}_Y^{RR-DM}(\pi)$ in eq. (8) | baseline/ablation |
| IPW | $\hat{V}_Y^{IPW}(\pi)$ in eq. (3) | $\hat{V}_Y^{RR-IPW}(\pi)$ in eq. (9) | baseline/ablation |
| DR | $\hat{V}_Y^{DR}(\pi)$ in eq. (4) | (see below) | Standard benchmark |
| Regression control variate | $\widehat{V}_Y^{RR\text{-}CV}(\phi(D;\pi))$ in eq. 11 | $\widehat{V}_{\hat{Y}}^{RR\text{-}CV}(\phi(\hat{D};\pi))$ in eq. 14 | Eq. 11: primary proposal; Eq. 14: denoised variant |

**Related work in multi-objective policy learning.** Several lines of research highlight the prevalence of multiple outcomes, although methodologically they are different. Viviano et al. (2021) and Ludwig et al. (2017) study multiple outcomes, but with a focus on hypothesis testing. Recent work investigates short-term surrogates or proxies vs. long-term outcomes (Athey et al., 2019; Knox et al., 2022), but this clearly designates one as a noisy version of the unavailable other. Instead, we focus on the policy learning setting with multiple outcomes in general, *without knowing which one is a proxy for the other.* Luckett et al. (2021) and Lizotte et al. (2012) both highlight methods that use expert-derived combined outcomes but say that these methods do not account for differences across units. Additionally, a complementary paper of Björkegren et al. (2022) leverages the many outcomes in an anti-poverty program and instead aims to uncover and audit what policy-makers value in allocation decisions.

Other works consider multi-objective policy learning via scalarization, but without denoising. Boominathan et al. (2020) study fully observed outcomes. Kennedy et al. (2019) consider effect estimation with mean-variance rescaling, but not optimal policies or further denoising. We focus on the benefits of dimensionality reduction for multiple outcomes. Other work uses factor models for policy learning with noisy outcomes. Chen et al. (2021) learn discrete-valued latent factors at baseline from a restricted Boltzmann machine model. Saito & Joachims (2022) consider (known) action embeddings for variance reduction. Our approach differs from previous work in estimating continuous constructs in the outcome space.

Factor models and low-rank matrix completion models have been of recent interest in the literature on synthetic control. Our setting is somewhat different: we assume unconfoundedness, unlike synthetic controls which posits factor structure on unobserved confounders. We focus on variance reduction for decision-making and we make an observable factor model assumption on outcomes, which is checkable. Although we used reduced rank regression, we can use other latent variable models as well. Another generative factor model is probabilistic PCA (Tipping & Bishop, 1999), but it implies an isotropic covariance structure that is incompatible with covariate-adjusted outcomes.[3] Other approaches include matrix generalizations of nuclear-norm regularization (Yuan et al., 2007), but this is computationally expensive.

## 3 Methodology

We propose a family of estimators that leverage the low-rank structure of multivariate outcomes. Our approach combines reduced rank regression with standard off-policy evaluation estimations to reduce variance when outcomes are high-dimensional. We first introduce a few ablations, reduced rank direct method (RR-DM) that imputes counterfactual outcomes using reduced rank regression, and denoised IPW (RR-IPW). Our last estimator family augments IPW with control variates to further reduce variance. The raw-outcome version preserves a clean control-variate/DR interpretation, while the denoised version is a more aggressive shrinkage variant that can improve finite-sample performance when the reduced-rank outcome model is accurate. Each estimator exploits the same low-rank structure from reduced-rank regression but trades off robustness and variance reduction.

---

[3]McKenzie (2005) even uses PCA on asset data to create an index variable as a measure of inequality, but due to the same issue the authors use the standard deviation of the first principal component instead.

**Direct Method with Reduced-Rank Regression (RR-DM).** The direct method estimates the conditional mean outcome using reduced rank regression and plugs this estimate in to the policy value function. We use our reduced rank estimation procedure to obtain estimators $\hat{A}_t$ and $\hat{B}_t$. Using RRR yields a more specialized estimator that leverages the structural low-rank assumption on the coefficient matrix. By contrast, a generic direct method that uses OLS separately for each outcome component of $Y$ yields a full-rank coefficient matrix; this serves as a natural baseline that our method improves upon.

The reduced-rank direct method estimator is

$$\hat{V}_Y^{RR-DM}(\pi) = \sum_t \mathbb{E}_n \left[ \pi(t \mid X) \rho^\top \hat{\mu}_t^{RR}(X) \right] \tag{8}$$

**Inverse Propensity Weighting with Reduced-Rank Regression (RR-IPW).** Denoised outcomes $\hat{Y}$, can also replace $Y$ in other estimators beyond the direct method. We introduce a variant of the IPW estimator that replaces raw outcomes with denoised predictions from the reduced-rank model:

$$\hat{V}_Y^{RR-IPW}(\pi) = \sum_t \mathbb{E}_n \left[ \pi(t \mid X) \frac{\mathbb{I}[T=t]}{\hat{e}_t(X)} \rho^\top \hat{\mu}_t^{RR}(X) \right] \tag{9}$$

Under Assumption 3, both of these estimators are unbiased. See Appendix B for the statement. However, for RR-IPW in particular, this can remove IPW's original advantage of being unbiased based on propensity scores $e(X)$ alone. It can be unintuitive to introduce this denoised IPW estimator, even as a potential ablation or baseline for our later variance-reduced control variate estimator. When could denoised IPW even improve upon IPW? If outcomes have large irreducible noise, and the reduced-rank regression $\hat{\mu}_t^{RR}$ has low prediction error relative to the true prediction function $\hat{\mu}_t$, then the MSE of denoised IPW could improve upon standard IPW - of course, at the cost of reduced model robustness and bias. Such tradeoffs put such an approach in the category of *shrinkage-based* approaches. Shrinkage of the importance sampling weights can improve variance of off-policy evaluation (Su et al., 2020). Our denoising approach improves MSE by the exact same mechanism; reducing the irreducible variance multiplying high-variance importance sample weights, at the cost of bias.

The next corollary formalizes this intuition and compares the mean-squared error of IPW vs. denoised IPW to isolate the bias-variance trade-off.

**Corollary 5** (A sufficient condition for arm-specific improvement from denoising). *Under the conditions of Proposition 15, assume that for some $L_\rho > 0$, $|\rho^\top Y| \le L_\rho$ a.s., $|\rho^\top \hat{\mu}_t^{RR}(X)| \le L_\rho$ a.s. Then a sufficient condition for improvement of the arm-t estimate, $MSE(\hat{V}_{n,t}^{RR-IPW}(\pi)) < MSE(\hat{V}_{n,t}^{IPW}(\pi))$, is equivalently, conditional on the fitted denoising function:*

$$\underbrace{\left( \mathbb{E}\left[ \pi(t \mid X) \left\{ \rho^\top \hat{\mu}_t^{RR}(X) - \rho^\top \mu_t(X) \right\} \right] \right)^2}_{\textit{squared bias from denoising}} + \frac{1}{n} \mathrm{Var}(\psi_t^{RR\text{-}IPW}(\pi)) < \frac{1}{n} \mathrm{Var}(\psi_t^{IPW}(\pi)).$$

*Expanding this out further,*

$$\left( \mathbb{E}\left[ \pi(t \mid X) \left\{ \rho^\top \hat{\mu}_t^{RR}(X) - \rho^\top \mu_t(X) \right\} \right] \right)^2 + \frac{2L_\rho}{n} \mathbb{E}\left[ \pi(t \mid X)^2 \left( \frac{1}{e_t(X)} - 1 \right) \left| \rho^\top \hat{\mu}_t^{RR}(X) - \rho^\top \mu_t(X) \right| \right]$$

$$< \frac{1}{n} \left\{ \mathbb{E}\left[ \pi(t \mid X)^2 \frac{\mathrm{Var}(\rho^\top Y \mid X, T=t)}{e_t(X)} \right] + \mathrm{Var}\left( \pi(t \mid X) \rho^\top \mu_t(X) \right) - \mathrm{Var}\left( \pi(t \mid X) \rho^\top \hat{\mu}_t^{RR}(X) \right) \right\}.$$

Thus, denoising helps when the weighted prediction error for $\mu_t$ is small relative to the overlap-weighted outcome variance, up to the variance correction term. The analogous DR comparison has the same form, with the outcome variance term replaced by the (irreducible) conditional residual variance $\mathrm{Var}(\rho^\top (Y - \mu_t(X)) \mid X, T=t)$.

We mention RR-DM and RR-IPW as potential baselines/ablations. Our primary suggested estimator is based on regression control variates, which we next introduce.

### 3.1 Deriving a control variate estimator

Although the direct method and IPW are unbiased under correct reduced-rank specification, potential drawbacks of these previous estimators are their reliance on the outcome model and the high variance that can be induced by inverse propensity weights. To address this, our final variant adds control variates to the IPW estimator, thereby weakening the reliance on model assumptions while reducing variance. The most robust version of this estimator begins with the original IPW estimator, although a more aggressively denoised version builds off of the RR-IPW estimator.

Control variate methods reduce variance by adding zero-mean terms that are correlated (or anti-correlated) with the random quantities being averaged, thereby reducing variance without introducing bias (Glynn & Szechtman, 2002). However, when the control variates are vector-valued, a natural question is, what is the optimal linear combination of control variates to minimize variance? To summarize our approach in answering this question and deriving a control variate estimator, we start by defining the control variate vector $C_t$, then we introduce the optimal weighting vector $D_t$. We find this optimal weighting vector by following the regression control variate approach introduced by (Glynn & Szechtman, 2002) and derive the functional form that minimizes the variance of our estimator. We show that our control variate estimator enjoys the unbiasedness properties of previous estimators as well as variance reduction because all weighted control variate vectors are also zero mean.

**Definition 6** (Outcome Control Variates). *For any multivariate function $h_t(X)$, the control variate vector is:*

$$C_t(h_t, X) = \left(1 - \tfrac{\mathbb{I}[T=t]}{\hat{e}_t(X)}\right) h_t(X),\tag{10}$$

*and for some weighting vector $D_t$, the control variate and variance-reduced IPW estimator are*

$$\phi_t^Y(D_t; \pi) = \pi(t \mid X)\left\{\frac{\mathbb{I}[T=t]\rho^\top Y}{\hat{e}_t(X)} + \hat{D}_t C_t\right\}$$

$$\hat{V}_Y^{RR-CV}(\phi(D; \pi)) = \sum_t \mathbb{E}_n[\phi_t^Y(D_t; \pi)].\tag{11}$$

Above, we add control variates to the IPW estimator using the observed outcomes $\rho^\top Y$ for model robustness. We can also replace $\rho^\top Y$ with denoised quantities $\rho^\top \hat{\mu}$ or $\rho^\top \hat{Z}$ for additional variance reduction.

To obtain a low-dimensional but informative control variate, we take $h_t(X) = \hat{B}_t X$, so that the resulting control variate vectors span the $r$-dimensional reduced-rank space (i.e., have rank $r$). Empirically, alternative choices of $h_t(X)$ do not materially change results (see Appendix), so we proceed with $h_t(X) = \hat{B}_t X$ for policy optimization. Thus the final outcome control variate vector that we use is $C_t = C_t(\hat{B}_t X, X)$.

Finally, *regression control variates* (Glynn & Szechtman, 2002) find a weight vector $D_t$ that achieves optimal variance reduction by maximizing correlation with the randomness in the original estimate. Such a weighting vector is:

$$D_t^* = (C_t^{*\top} C_t^*)^{-1} C_t^{*\top}(\rho^\top Y),\tag{12}$$

where $C_t^*$ is defined in terms of the true $B_t$ and $e_t(X)$ or $C_t^* = (1 - \tfrac{\mathbb{I}[T=t]}{e_t(X)})B_t X$. We have that $D_t$ is the pseudoinverse solution to $\rho^\top Y = C_t D_t$. It corresponds to the regression control variate. For our estimated control variate variants, we introduce the estimated weighting vector

$$\hat{D}_t = (C_t^\top C_t)^{-1} C_t^\top(\rho^\top \hat{\mu}).\tag{13}$$

A denoised version of the control-variate estimator replaces $Y$ with the reduced-rank regression estimate:

$$\hat{V}_{\hat{Y}}^{RR-CV}(\phi(\hat{D}; \pi)) = \sum_t \mathbb{E}_n[\phi_t^{AZ}(\hat{D}_t; \pi)] = \sum_t \mathbb{E}_n\left[\pi(t|X)\left\{\frac{\mathbb{I}[T=t]\rho^\top \hat{A}\hat{Z}}{\hat{e}_t(X)} + \hat{D}_t C_t\right\}\right],\tag{14}$$

$$C_t = \left(1 - \tfrac{\mathbb{I}[T=t]}{\hat{e}_t(X)}\right) \hat{B}_t X, \quad \hat{D}_t = (C_t^\top C_t)^{-1} C_t^\top(\rho^\top \hat{\mu}).$$

Although we use *estimates* of $Y$ or $Z$, we show consistency for the regression control variate.

Importantly, the standard doubly-robust estimator is within the class of control variates that the regression control variate (eq. (11), on original outcomes) implicitly optimizes over. And, we know from efficiency theory that the doubly-robust estimator is optimal. Therefore, when the DR control variate lies in the candidate span, the raw-outcome regression control variate in eq. (11) optimizes over a class that includes the DR choice and can reduce conditional empirical variance relative to that choice in finite samples.

However, the denoised control variate estimator (eq. (14)), which additionally replaces the observed $\rho^\top Y$ with the denoised prediction $\rho^\top \hat{\mu}_t^{RR}(X)$, does not enjoy this connection to the standard doubly-robust estimator. We view the denoised control variate estimator as a shrinkage estimator whose potential gains reflect finite-sample bias–variance tradeoffs, without the asymptotic guarantees of uniform weak improvement over standard DR.

The next results show that the feasible variants still enjoy variance reduction properties (stated below in terms of $Y$).

**Lemma 7** (Unbiasedness of CV Estimator). *Under Assumption 3, the policy value with the control variate estimator (for $\hat{Z}$ and $\hat{Y}$) is unbiased:*

$$\mathbb{E}\left[\sum_t \mathbb{E}_n[\pi(t \mid X)\phi_t^Y(D_t;\pi)]\right] = V_Y(\pi).$$

**Proposition 8** (Consistency in OLS with Noisy Outcomes). *Define the oracle weighting vector $D_t^* := (C_t^\top C_t)^{-1} C_t^\top (\rho^\top Y_t)$. For $t \in \{0,1\}$,*

$$\hat{D}_t \xrightarrow{p} D_t^*. \tag{15}$$

We abbreviate $\phi_t(D_t) = \phi_t^Z(X,T,Y;D_t)$ when it's clear from context; the following results hold for $\phi_t^Z(X,T,Y;D_t)$ or $\phi_t^Y(X,T,Y;D_t)$. Proposition 8 shows that the estimated control variate is asymptotically consistent. Next, we show that the control variate estimator achieves the optimal variance asymptotically.

**Theorem 9.** *Assume that $\mathbb{E}\left[CC^\top\right]$ is nonsingular and $\mathbb{E}\left[\rho^\top \hat{Z}\hat{Z}^\top \rho + C^\top C\right] < \infty$, where $\hat{Z} = \hat{B}_t X$. Suppose $\mathbb{E}[C] = 0$ and that $\hat{D}_t \rightarrow_p D_t^*$ as $n \rightarrow \infty$. Then, for $t \in \{0,1\}$, as $n \rightarrow \infty$, we have that $n^{1/2}(\hat{V}^{RR-CV}(\phi(\hat{D};\pi)) - \hat{V}(\phi(D^*;\pi))) \rightarrow_p 0$ and therefore:*

$$n^{1/2}(\hat{V}^{RR-CV}(\phi(\hat{D};\pi)) - V_Y(\pi)) \Rightarrow N(0, \text{Var}[\phi(D^*;\pi)]).$$

Theorem 9 shows that our denoised variants of the control variate estimator are asymptotically consistent for the analogous variance-optimal estimator. It asserts that our control variate estimate $\hat{D}$ is consistent for $D^*$. Lastly, Algorithm 1 provides pseudo-code for the procedure for learning the direct method estimator with the reduced rank models. A similar procedure can be applied to the other derived estimators.

### 3.2 Analysis

We next derive finite-sample generalization bounds for the out-of-sample policy value. We assume the policy class is finite, i.e. $|\Pi| = K$, although this can easily be generalized to infinite policy classes with standard use of covering numbers. We make some standard assumptions regarding bounded outcomes and known propensity scores.

**Assumption 10** (Bounded outcomes). *For any $X, T$, the outcomes $|Y_i|_\infty \leq L$ almost surely.*

**Assumption 11** (Overlap in propensity scores). *For any $x, t$, we have that $0 < e_t(x) < 1$.*

---

**Algorithm 1** Noise-reduced RR Direct Method

---

1: Input: data $(X, T, Y)$, $\rho$, $\Theta$ parameter space for policy $\pi_\theta$
2: Standardize the data and outcomes, e.g. demean the data and standardize $X$ to isotropic covariance and divide $Y$ by the standard deviation.
3: Obtain $\hat{e}_t(X)$ by regressing $T \sim X$.
4: Obtain $\hat{\mu}_t^{RR}(X)$ by estimating reduced-rank models for $t = 0, 1$.
5: Policy learning: Initialize policy $\pi_\theta$
6: **for** t=1,2,$\cdots$ **do**
7: $\quad \theta^{(t)} = \theta^{(t-1)} - \eta \cdot \nabla_\theta V_Y(\pi_\theta^{(t-1)})$
8: **end for**

---

**Theorem 12.** *Suppose Assumption 10, Assumption 11, and that the propensity score is known or conditionally unbiased. With probability $1 - \delta$, we have that*

$$V_Y(\pi^*) - V_Y^{RR-CV}(\hat{\pi}_n) \leq \sqrt{2\log\left(\frac{2K}{\delta}\right) \cdot \frac{\sup_{\pi \in \Pi} \text{Var}[\phi(D^*; \pi)]}{n}} + \frac{2L}{3n}\log\left(\frac{2K}{\delta}\right) + O_p(n^{-\frac{1}{2}})$$

The generalization bound depends on the (worst-case) variance of the estimator and therefore illustrates how control-variate estimators improve generalization. The second estimator-independent term depends on the prediction error of estimating $\hat{Z}$. Analogous results for $\hat{Y}$ are provided in the Appendix. Additionally, in Theorem 12 and Theorem 21, we assume that the propensity scores are known, or if estimated, are conditionally unbiased. We provide Corollary 22 in the appendix that discusses what happens to the generalization bound when propensity scores are estimated in general.

## 4 Experiments

We demonstrate the effectiveness of our approach using synthetic simulations and a real-world dataset from a randomized controlled trial involving cash transfers, a setting characterized by many observed outcomes. Because many real-world outcome vectors are redundant/noisy measures of underlying constructs, we first validate in both synthetic and real data that low-rank structure exists and can be recovered.

### 4.1 Simulated data

We demonstrate the performance of our estimators (introducing additional ablations and baselines) on simulated data with known ground-truth. The data generation process for these experiments follows the reduced-rank latent variable model described in Assumption 3. We generate the dataset as follows: we generate $p$-dimensional $X_i \overset{iid}{\sim} N(M_p, I_p)$ for $i = 1, ..., n$, where $M_p$ is the randomly generated mean matrix. Given $X$, the treatment assignment $T_i$ is a Bernoulli with logistic treatment probability $P(T_i = 1 | X_i = x) = e^{\beta^\top x}/(1 + e^{\beta^\top x})$. Next we generate $B_t$, a $p \times r$ matrix, and $A_t$, a $k \times r$ matrix with independent, standard normally-distributed entries. We set $r = 2, k = 5, p = 8$. We run reduced rank regression on the observed outcomes $Y$ to get estimators $\hat{A}_t, \hat{B}_t$.

**Latent Outcome Estimation Reduces Variance in Off-Policy Evaluation** Overall, our comparisons illustrate the benefit of denoising noisy outcomes $Y$ for reducing the variance of off-policy evaluation. We compare the variance reduction of our estimators: direct, IPW, and DR/CV. We report the variance reduction achieved from estimating $Z$ and using control variates. Since practitioners can't compute estimators based on true latent outcomes $Z$, we focus on improvements of the feasible estimator using estimated latent outcomes, $\rho \hat{A}_t \hat{Z}_t$. The hypothetical situation is that a decision-maker can specify a weighting/trade-off vector $\rho \in \mathbb{R}^k$: our methods can denoise using predictions of $\hat{Y}$, and we can compare the improvement of our method vs. feasible estimators that use the original noisy $Y$ outcomes. In the following experiments, we set $\rho = [0.3987, 0.0212, -0.6195, 1.3661, -1.593]$. This weighting vector parameter $\rho$ is randomly drawn from

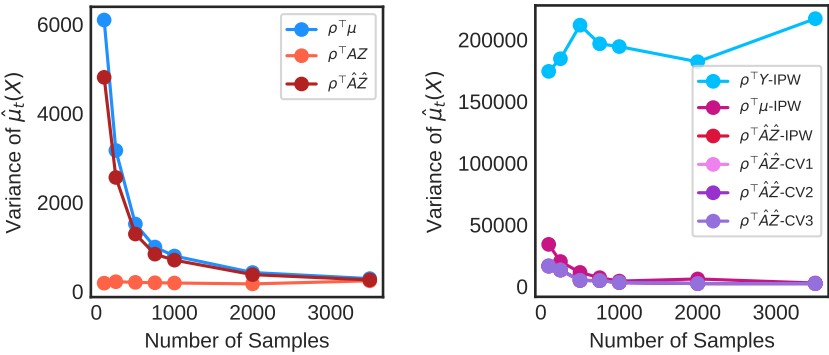

Figure 1: Variance in ATE estimation. **Left:** Comparison of variances averaged over 100 datasets as the sample size of the dataset increases using the direct estimators for $\rho^\top Y$. **Right:** Comparison of variances as the sample size increases for the IPW and control variate estimators. Note that all the $\hat{A}\hat{Z}$-estimator variants perform similarly and are stacked on top of each other in the plot. *Lower is better.*

a normal distribution. Here, we must emphasize that our choice of $\rho$ is meant to emulate what is done in practice, but the utility of our denoising approach exists regardless of the choice of $\rho$. In practice, experts will specify this weighting vector to bring together important variables for their target construct. The focus of this work is not on the choice of $\rho$ and in some sense, this choice is arbitrary for our proposed methodology.

**Baselines and ablations**. To thoroughly compare the performance of our methods, we compare our estimators to the oracle denoised outcomes $\rho^\top AZ$ and we introduce additional ablations and baselines. First, we introduce a set of ablations that highlight the essential improvements from dimensionality reduction, by ablating the RRR regression model (satisfying Assumption 1) vs. full-rank linear regression (OLS):

- $\rho^\top \mu$ is the direct method with OLS (full-rank regression coefficient matrix),

- $\rho^\top \mu$-IPW which is IPW replacing $\rho^\top Y$ with $\rho^\top \mu$ (OLS; full-rank coefficients),

- and $\rho^\top \mu$-CV which is our control variate estimator with $Y$-estimation $\mu$ with OLS rather than our RRR estimates.

Next we consider baselines with the original noisy $Y$ observations:

- $\rho^\top Y$-IPW is standard IPW (eq. (3)),

- $\rho^\top Y$-DR is the DR estimator with OLS (eq. (4)).

Next we introduce our proposed methods:

- $\rho^\top \hat{A}\hat{Z}$ is the direct method with RRR (eq. (8)),

- $\rho^\top \hat{A}\hat{Z}$-IPW replaces observed outcomes with denoised (eq. (9)),

- and $\rho^\top \hat{A}\hat{Z}$-CV is our derived CV estimator (eq. (14)).

Later in Figure 2 we compare all of these methods and in Figure 3 we compare each estimator variant to its most relevant baselines and ablations. Figure 1 compares the variance of all the estimators for the ATE as the sample size increases. We observe significant variance reduction due to denoising $Y$, which diminishes further benefit of using the control variates. In the appendix, we include additional experiments to assess how changing relative dimension of $Y$ and $Z$ and the observation noise level of $Y$ affects variance reduction.

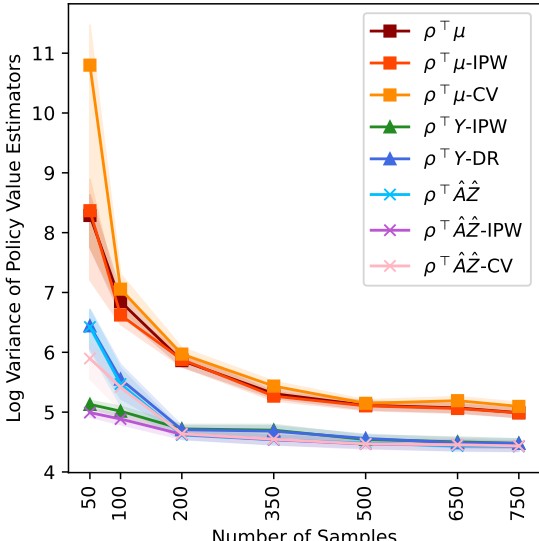

Figure 2: Policy evaluation experiment: comparing variances of the policy value for each estimator averaged over 1000 datasets. We compute each policy value by using the optimal policies learned under the true latent outcomes $\rho^\top A_t Z$. *Lower is better.*

Our figures show that the ablations $\rho^\top \mu, \rho^\top \mu-\text{IPW}, \rho^\top \mu\text{-CV}$ perform worse than their RRR counterparts, indicating that it is our structural assumption of *lower-dimensional constructs* that leads to significant variance reduction rather than our parametric linear model specification.

**Policy evaluation.** For policy evaluation, in Figure 2 we compare all of the estimators evaluated on an estimate of the oracle-optimal policy $\pi^*$ (i.e., obtained by running policy optimization on the true latent outcomes $\rho^\top A_t Z$). We evaluate the variance reduction achieved in the objective value and policy value and use the optimal policy $\pi^*$. To obtain $\pi^*$, we run the optimization procedure with subgradient descent for 20 iterations. We evaluate the estimators over 1000 trials. In Figure 2, $\rho^\top \hat{A}_t \hat{Z}$ improves especially when the sample size is small. We see the benefits of denoising when compared to our ablations and even the doubly robust estimator. We also observe that for small $n \leq 100$, the IPW-family of estimators (with denoising and without) likely before best because there is no extra nuisance estimation. In higher sample regimes, when there is enough data to estimate an outcome model reasonably, the DR and CV estimators perform just as well.

Notably, for every estimator, the denoised version does better. We recommend standardizing outcomes (i.e. so that the simple average is the precision-weighted average), and recall that IPW can be thought of as a doubly-robust estimator imputing 0 for counterfactual outcomes. Although in the simulated data, we did not standardize outcomes, the outcomes are zero-mean in the data-generating process. Without standardization of the features and the outcomes, our methods suffer from the common issues seen with regularized regression and gradient descent optimization where larger values shrink disproportionally compared to smaller values of data. In this setting, combining the denoised $\hat{Y}$ estimate with IPW does well and is similar to the CV-based estimator: we think this is because IPW with standardized outcomes conducts additional shrinkage towards the mean.

**Evaluating multi-objective policy optimization**. Lastly, in Figure 3 we run policy optimization with all of the estimators where we use a subgradient method to carry out the policy optimization. We run the optimization procedure over 40 iterations for each of the 50 trials. In the left column of Figure 3, we take the learned policy $\hat{\pi}$ from each of the estimation methods and evaluate the variance of the policy value on an out-of-sample dataset. In the right column of Figure 3, we compare recovery of the optimal policy via the log mean-squared error difference in policy value of $\hat{\pi}$, $V(\hat{\pi})$ and the policy value of a pre-computed optimal policy $\pi^*$, $V(\pi^*)$. The first row of figures ablates the improvements from our latent variable model

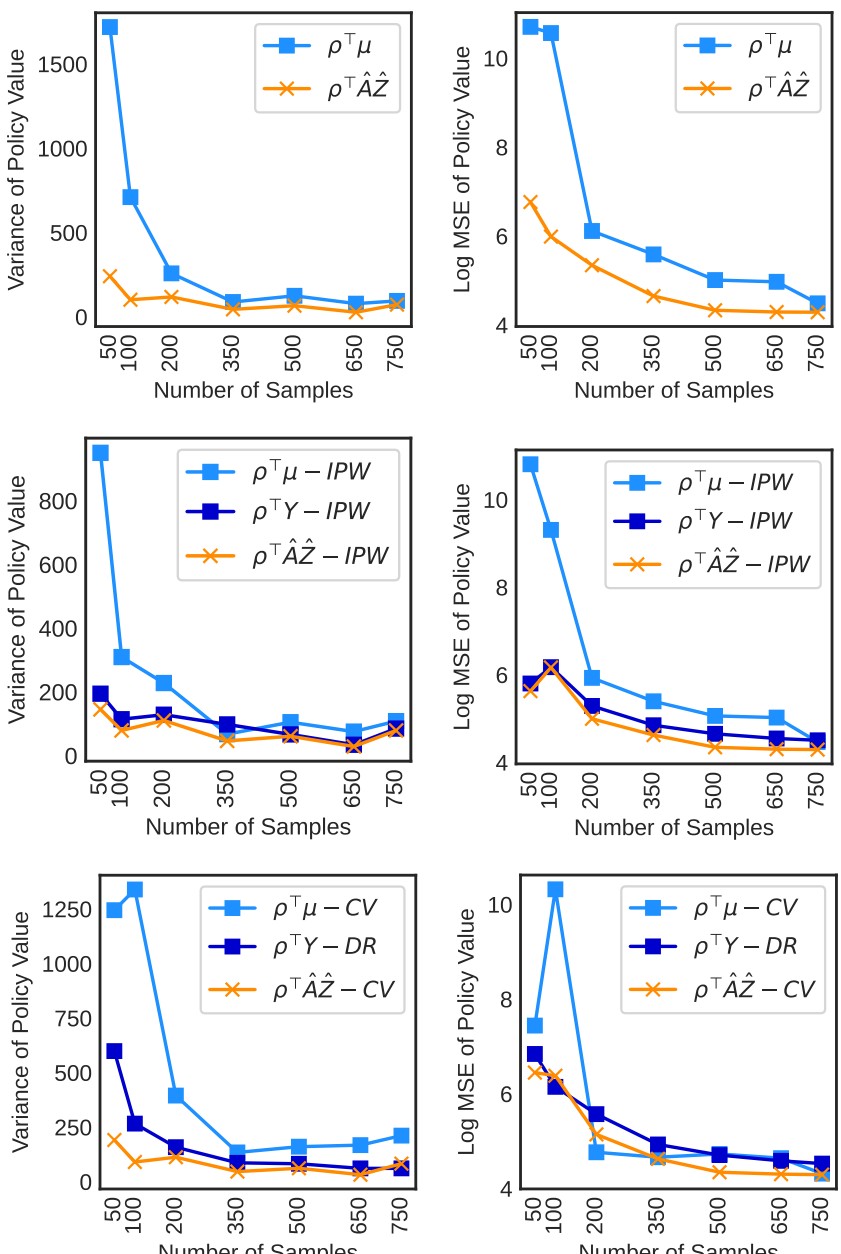

Figure 3: Policy optimization experiment: left figures illustrate variance of out-of-sample policy value estimate (averaged over 50 datasets). Right figures compare the log MSE for policy value suboptimality. *Lower is better.* First pair of figures compares $\rho^\top \hat{A}\hat{Z}$ (reduced rank DM) with full-rank DM. Second pair of figures compares standard IPW with our denoised IPW, and last pair compares standard doubly robust estimator ($\rho^\top Y$-DR) with our control variate estimator.

in the direct method. The second row of figures compares standard IPW with our denoised IPW. The last row of figures compares the standard doubly robust estimator with our RRR-based and control variate estimator (our preferred specification). **Our method (in yellow, $\rho^\top \hat{A}\hat{Z} - CV$) does best, and in fact, all denoised variants achieve variance reduction that translates directly into improved learned policy performance.**

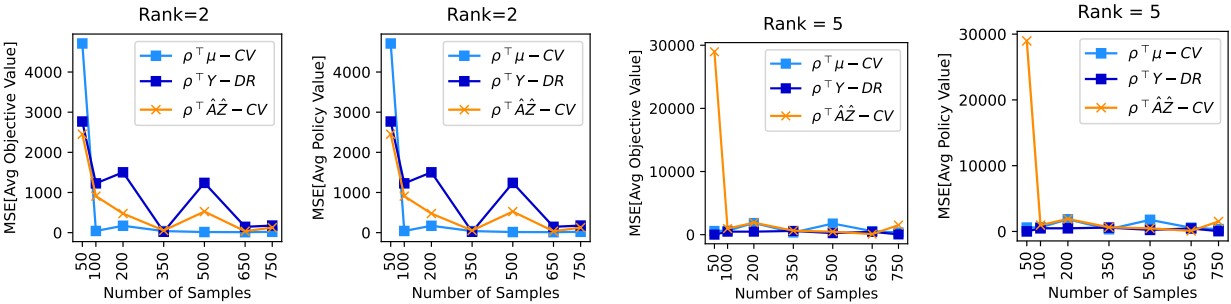

Figure 4: Robustness to nonlinearity policy optimization experiment: figures illustrate the MSE of the averaged objective value and averaged policy value (averaged over 50 datasets). Left figures make a low rank (rank=2) nonlinear assumption and right figures make a moderate rank (rank=5) nonlinear assumption. *Lower is better.*

**Robustness to Nonlinearity.** We additionally evaluate the robustness of our reduced-rank approach under violations of linearity and in comparison to flexible nonlinear models. Although reduced-rank regression is a linear model, many real-world datasets exhibit nonlinear structure in both the covariates and the outcome relationships. We develop robustness checks for our method in a set of simulations where the true outcome model includes nonlinear transformations of both $X$ and $Z$ (here a *sine* function), and low and moderate underlying rank assumptions. We follow a similar data generating process to what we defined above, except now we change the underlying rank to be set to $r = 2$ for the low rank assumption and we set $r = 5$ for the moderate rank assumption. We generate the matrices $B_t$ and $A_t$ and set Z as $Z(t) = B_t X + \sin(X) + U$ and Y as $Y(t) = A_t Z(t) + \sin(Z(t)) + \epsilon$ where $U$ and $\epsilon$ are generated from standard normal distributions. We compare to nonlinear baselines with a flexible nonparametric outcome regression model (i.e. random forest regression), which can capture complex interactions and nonlinearities. Figure 4 plots the performance of our control variate estimator ($\rho^\top \hat{A}\hat{Z}$-CV, yellow) against two other baseline estimators, the full-rank estimator with control variates ($\rho^\top \mu$-CV, light blue) and the DR estimator ($\rho^\top Y - DR$, dark blue). Across these experiments, our method continues to perform well — *improving upon* or nearly matching the DR estimator with black-box ML, otherwise the standard recommendation in causal ML. The reduced-rank structure provides a useful modeling approach in the case of multiple outcomes, and by design, our estimator is relatively robust to moderate departures from linearity.

Our experimental setup strives to assess the improvements from both dimensionality reduction and our proposed variance reduction estimator. To do this, we compare against a set of ablations that highlight the benefits of dimension reduction and another set of baselines to show to improvements from our estimation procedure with variance reduction. Overall, our comparisons highlight the benefit of our denoising approach for multiple noisy outcomes $Y$ in policy evaluation and optimization. While our method assumes a latent linear model, the benefits achieved are robust to deviations in this assumption. Our results show that denoising the outcomes improves policy evaluation, while our variance reducing estimators perform similarly to their baselines suggesting that denoising is driving the improved results. For policy optimization, we see that denoising improves results across the board. Additionally our control variate estimator plus denoised $Y$ improves policy optimization performance the most over the other ablations with no variance reduction ($\rho^\top Y - DR$) and no denoising ($\rho^\top \mu - CV$).

Importantly, when the low-rank assumption fails such as when the true rank is high or the outcome relationships are highly nonlinear, then our method degrades gracefully and trades off some model bias for variance reduction, as the trade-off analysis of Corollary 5 indicates. In such cases, the variance reduction from denoised outcomes via the reduced-rank model can still yield competitive results in terms of mean squared error. Crucially, the low-rank assumption is well-motivated by the substantive structure of the social data settings we aim to target. In policy evaluation and optimization with many social outcomes, it is common for high-dimensional outcome vectors to be driven by a small number of latent factors, whether

due to correlated behavioral responses or common underlying social constructs. Reduced-rank regression is precisely designed to exploit this structure, and our method is tailored to settings where this low-dimensional signal is present but obscured by noise and measurement error.

To make this more concrete, let us revisit a generic multiple-outcomes setting of a large development randomized-controlled trial, discussed in the introduction and our later real-world data. Low-rank assumptions may be true for a portfolio of outcomes based on consumption (as broken down into various categories) savings, income and investments: these are all plausibly explained as linear combinations of a few factors: income, expenses, and overall consumption. Low-rank assumptions are less likely true when a portfolio of outcomes includes outcomes that weakly depend on everything else. For example, suppose multiple outcomes include a comprehensive set of 1-year endline surveyed outcomes spanning well-being/health/employment/education/income and longer-term 5-year income measures measured in administrative data: long-term 5-year income likely depends weakly on all such 1-year endline outcomes, and therefore low-rank is an *approximation*, not the underlying truth. However, if long-term 5-year income depends much more on a few factors like earlier employment/education/income, as the surrogate index literature argues (Athey et al., 2019), low-rank may be a useful variance-reducing approximation.

Table 2: Out-of-sample variance for DM, IPW, DR, and CV estimators under policy evaluation (**lower** is better). The variance is much lower with our proposed denoising approach $\mu^{RR-DM}$ and $\mu^{RR-IPW}$ and our control variate approaches $\mu^{CV}$ and $\mu^{RR-CV}$.

| Estimator | $\rho^\top \mu^{DM}$ | $\rho^\top \mu^{RR-DM}$ | $\rho^\top Y^{IPW}$ | $\rho^\top \mu^{IPW}$ | $\rho^\top \mu^{RR-IPW}$ | $\rho^\top Y^{DR}$ | $\rho^\top \mu^{CV}$ | $\rho^\top \mu^{RR-CV}$ |
|---|---|---|---|---|---|---|---|---|
| Out-of-Sample Variance | 4.373 | 4.373 | 111.746 | 8.369 | 5.009 | 111.236 | 2.757 | 5.058 |

## 4.2 Real world case study: "Sahel" dataset, poverty graduation program.

We turn to a dataset derived from a multi-country randomized control trial that evaluated social safety net programs aimed at graduating households out of poverty (Bossuroy et al., 2022a). The program duration spanned 2 to 5 years and the data was collected in four countries including Niger, Senegal, Mauritania, and Burkina Faso. The original dataset consisted of 8,779 households and 401 features, but the final data after data cleaning and preprocessing consists of 4,139 households and 36 features. Additionally, we selected *seven* outcomes (direct program targets), screened for treatment effect heterogeneity. These include beneficiary wages, food consumption, business health, beneficiary productive revenue, business asset value, investments, and savings. Note unlike our previous generic assumption on cost outcomes, these are benefits that we would like to *maximize*.

### 4.2.1 Experimental setup

**Features.** The dataset contains baseline characteristics. The survey used to collect this information included questions about productive and non-productive household activities, housing, assets, familial relationships, psychology and mental health, food consumption spending, education spending, and health spending. The final dataset used for experimentation consists 4,139 households and 36 features.

**Treatments.** Although the original experiment considered multiple treatment levels that are different combinations of six previously-identified financial or psycho-social interventions, we focus on a subset of treatments: control arm (no intervention) and full treatment (received all six interventions). See the appendix for more details.

**Outcomes.** Program effects were measured not only with two primary outcomes (food insecurity and consumption per capita), but also a large set of secondary outcomes. These secondary outcomes include those that are directly targeted by the program, downstream outcomes not directly targeted, and other descriptive measures. We selected *seven* outcomes (direct program targets), screened for treatment effect heterogeneity. These include beneficiary wages, food consumption, business health, beneficiary productive revenue, business asset value, investments, and savings. Note unlike our previous generic assumption on cost outcomes, these are benefits that we would like to *maximize*.

Table 3: Out-of-sample evaluation of optimized policy value (**higher** is better). Rows indicate policies optimizing different estimates on the training set. Columns indicate different evaluation methods on the test set. Estimated factors $\hat{Z}$ have dimensions equal to 4. We report the mean ± standard deviation. Bold is best (within-column), italic second-best.

| $\pi^*$ | Self | $\rho^\top \mu^{RR-DM}$ | $\rho^\top Y^{IPW}$ | $\rho^\top Y^{DR}$ |
|---|---|---|---|---|
| $\rho^\top \mu^{DM}$ | $0.101 \pm 1.836$ | $0.085 \pm 1.582$ | $\mathit{0.180} \pm 11.959$ | $\mathit{0.148} \pm 11.835$ |
| $\rho^\top \mu^{RR-DM}$ | $0.108 \pm 1.618$ | $\mathbf{0.108} \pm 1.618$ | $\mathit{0.180} \pm 12.166$ | $0.145 \pm 11.994$ |
| $\rho^\top Y^{IPW}$ | $-0.224 \pm 10.120$ | $-0.304 \pm 1.691$ | $-0.224 \pm 10.120$ | $-0.216 \pm 10.100$ |
| $\rho^\top \mu^{RR-IPW}$ | $0.124 \pm 2.461$ | $\mathit{0.103} \pm 1.628$ | $\mathbf{0.211} \pm 12.313$ | $\mathbf{0.190} \pm 12.143$ |
| $\rho^\top Y^{DR}$ | $-0.056 \pm 10.149$ | $-0.265 \pm 1.710$ | $-0.077 \pm 10.228$ | $-0.056 \pm 10.149$ |
| $\rho^\top \mu^{RR-CV}$ | $0.124 \pm 2.468$ | $\mathit{0.103} \pm 1.628$ | $\mathbf{0.211} \pm 12.313$ | $\mathbf{0.190} \pm 12.144$ |

**Preprocessing.** We split into training (3,311 households, for optimizing policies) and test datasets (828 households, for out-of-sample off-policy evaluation) and standardize (i.e. subtract the mean and divide by the standard deviation) for both outcomes $Y$ and features $X$.

**Model selection: Choosing the RRR model rank.** In the real data experiments, we do not know the lower rank of the reduced-rank regression a priori. We illustrate a method to choose what rank to use in the RRR, akin to an "elbow" plot used in other settings when choosing a lower rank for approximation. We propose calibration by the following procedure: vary RRR estimation over a range of candidate ranks, and consider bias-variance trade-offs in the fraction of outcome variance explained. For each rank $r$, we fit a reduced-rank regression model on the training set and compute the variance explained on both the training and held-out test sets, where a weighted variance explained is computed as an average of the $R^2$ coefficient for each dimension of $Y$ weighted by each outcome's marginal variance. The weighted variance explained is formally defined as

$$\text{VarianceExplained} = \frac{\sum_{j=1}^{k} \hat{\sigma}_j \cdot R_j^2}{\sum_{j=1}^{k} \hat{\sigma}_j}, \tag{16}$$

where $\hat{\sigma}_j = \sum_{i=1}^{n}(y_{ij} - \bar{y}_{ik})^2$ and $R^2 = 1 - \frac{\sum_i (y_{ik} - \hat{\mu}_{ij}^{r,RR}(X_i))^2}{\hat{\sigma}_j}$. Figure 5 plots these quantities as a function of the rank. The training variance explained increases monotonically with rank. In contrast, the test variance explained increases slightly for small ranks, peaks at a modest rank (here around $r = 2$ to $r = 4$), and then declines as the model overfits. Because the true underlying rank is unknown in practice, for our model selection we will rely on held-out performance rather than training fit. The test-set curve therefore provides empirical evidence for choosing a small rank that achieves the best predictive performance. While the absolute variance is generally low for these noisy real world outcomes, this result highlights that the reduced-rank structure can be beneficial for regularization, and that over-parameterizing the rank leads to worse out-of-sample generalization even though it improves in-sample performance.

Based on this analysis, our practical recommendation for rank selection is to select the rank that minimizes the estimated policy value. In our model selection experiments, we also see that this choice is aligned with the rank that maximizes the weighted variance explained on the validation set. The rank achieving the results in Table 3 both maximizes the policy value and explains the most variance in outcomes - suggesting these two ways of rank selection may align well. The rank we chose also achieves the best out-of-sample performance on the evaluation set. However, even if we had chosen a different value of the rank, our results in Tables 6 to 9 show that our method has the second-best performance, suggesting some robustness to the exact choice of rank, and general benefits of variance reduction.

### 4.2.2 Results

**Policy evaluation.** We assess variance reduction for policy evaluation. We (arbitrarily) evaluate a near-optimal policy: we obtain $\hat{\pi}_{DM}^*$, by optimizing a naive direct method estimator with OLS. We set $\rho = [8.28, 1.31, 0.21, 0.061, 0.59, 0.01, 1.12]$.

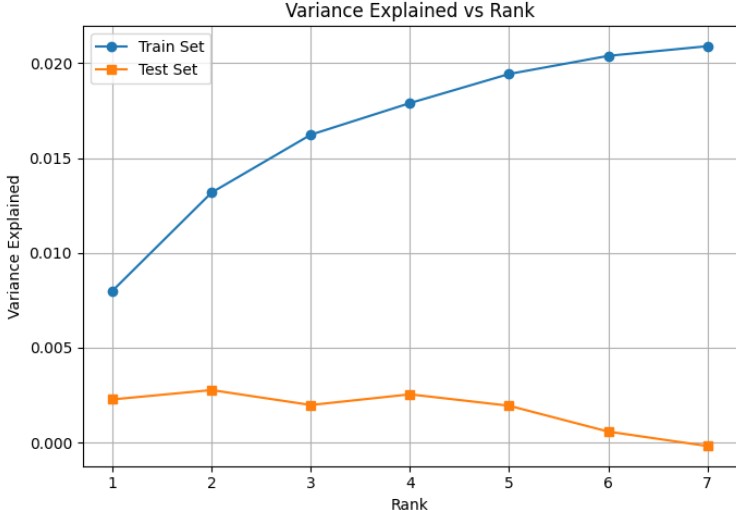

Figure 5: Variance explained by rank: We compute the variance explained at each reduced-rank model rank value. The plot shows the weighted variance explained on the training and test sets for reduced-rank regression models fit at ranks $r = 1, \ldots, 7$. Training variance explained increases monotonically with rank, while Test-set performance peaks at a low rank before declining. This plot validates our low-rank assumption in this real data setting and suggests using a rank of $r = 2$ or $r = 4$ for optimal performance.

Given that we already used the training dataset to find a near-optimal policy, we assess the out-of-sample variance on the test set. In Table 2, we see that the variance is much lower with our proposed denoising approach $\mu^{RR-DM}$ and $\mu^{RR-IPW}$ and our control variate approaches $\mu^{CV}$ and $\mu^{RR-CV}$. Although the bias remains unknown in real data, we obtain evidence of overall improvement in the next policy optimization experiment.

**Policy optimization.** Ultimately we would like to learn a treatment policy that *maximizes* these financial outcomes for each household. We learn optimal policies using the range of methods described previously in "Baselines and Ablations." Since this is a real dataset, we don't have the true counterfactual outcomes for observed individuals, but we can again use off-policy evaluation to *estimate* the value of the learned policies. In this real-world data experiment, where we do not know the true underlying data generating process, we ensure the best possible estimation of $\hat{Z}$ by conducting a hyperparameter sweep for our reduced rank regression estimates (see appendix for details). We set $\rho = [8.28, 1.31, 0.21, 0.061, 0.59, 0.01, 1.12]$.

In Table 3, we provide the out-of-sample estimates of the policy value using a series of different policies $\hat{\pi}^*$: each row optimizes a different policy value estimator. The best that we can do is evaluate on the held-out test set. Columns indicate different evaluation methods: first we evaluate each policy against its own estimation method, and then with the model-robust IPW and DR methods. The last two columns provide model-free evidence of our $\hat{Z}$ estimation procedure being robust to model misspecification. Across all the evaluation techniques (table columns), we see consistent improvement in the policy value and the within-column orderings of each evaluation is relatively stable. To summarize, our methods ($\rho^\top \mu^{RR}$ with DM, IPW, or CV) are consistently better than no denoising at all. Our denoised method has counterintuitive practical benefit in such social data settings, where common folklore is that causal effects in the social sciences are very noisy zeros and many objectives are noisy realizations of some underlying notion. Empirically, we see the variance reduction and improved causal estimation from our proposed method which ultimately leads to improved policy learning. From our learned treatment policies, we see that the following features: (i) whether the household head is female, (ii) household head age, (iii) number of years of education, and (iv) difficulty rating for lifting 10kg bag are the most important features that result in being treated by the policy.

## 5    Conclusion

We highlighted how multiple outcomes can pose estimation challenges, and developed tools using dimensionality reduction in multivariate regression to reduce variance and improve policy evaluation and optimization. Directions for future work include reducing model dependence via standard reweighted maximum likelihood estimation for the outcome model, or more advanced nonlinear dimensionality reduction, and employing model selection.

## 6    Ethical Statement

Our work is methodological in nature but we in particular focus on social settings, so care is warranted. Latent variables in particular have a history of being used in the social sciences in overly reductive or essentialist ways, which we caution against in general. Our focus here is on variance reduction, perhaps even given already-existing broad categorizations like "financial well-being" and "educational outcomes", but we recommend additional use of interpretive tools in practice to interpret and validate latent factors.

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

## Appendix

## A  Additional discussion

### A.1  Related work

**Other factor models.** Although we focus on reduced rank regression, we briefly describe some advantages related to other possible latent factor models, some of which could be equivalently adapted. Another probabilistic latent variable is probabilistic principal component analysis (PPCA) (Tipping & Bishop, 1999). A common latent variable model assumes a linear relationship between latent and observed variables; PPCA assumes that, conditional on the latent variables, the observed data Y are Gaussian. The marginal distribution over the latent variables is therefore *isotropic* Gaussian, and the implication of equal variance is a poor fit for heterogenous covariate-adjusted outcomes. Another option is a penalized least squares factor model. (Yuan et al., 2007) proposes simultaneous estimation of factors and factor loadings with a sparsity penalty, generalizing ridge regression for factor estimation. Unfortunately they penalize the coefficient matrix's Ky Fan norm, which is computationally intensive and overestimates the rank.

## B    Deferred Results

**Lemma 13** (Unbiasedness of DM Estimator). *Under Assumption 3, the policy value with the direct method estimator (for $\hat{Z}$ and $\hat{Y}$) is unbiased:*

$$\mathbb{E}\left[\sum_t \mathbb{E}_n\left[\pi(t \mid X)\rho^\top \hat{\mu}_t^{RR}(X)\right]\right] = V_Y(\pi)$$

unbiased.

**Lemma 14** (Unbiasedness of IPW Estimator). *Under Assumption 3, the policy value with the IPW estimator (for $\hat{Z}$ and $\hat{Y}$) is unbiased:*

$$\mathbb{E}\left[\sum_t \mathbb{E}_n\left[\pi(t \mid X)\frac{\mathbb{I}[T = t]\rho^\top \hat{\mu}_t^{RR}(X)}{e_t(X)}\right]\right] = V_Y(\pi).$$

**Proposition 15** (Arm-specific MSE comparison for RR-IPW). *Fix $t \in \{0,1\}$. Let*

$$m_t(X) := \rho^\top \hat{\mu}_t^{RR}(X), \qquad \mu_t^\rho(X) := \rho^\top \mu_t(X), \qquad \sigma_{\rho,t}^2(X) := \mathrm{Var}(\rho^\top Y \mid X, T = t).$$

*Assume $m_t$ and $e_t$ are treated as fixed functions (for example, estimated on an independent sample). Define*

$$\hat{V}_{n,t}^{IPW}(\pi) = \frac{1}{n}\sum_{i=1}^n \pi(t \mid X_i)\frac{\mathbf{1}\{T_i = t\}\rho^\top Y_i}{e_t(X_i)},$$

*and*

$$\hat{V}_{n,t}^{RR-IPW}(\pi) = \frac{1}{n}\sum_{i=1}^n \pi(t \mid X_i)\frac{\mathbf{1}\{T_i = t\}m_t(X_i)}{e_t(X_i)}.$$

*Let*

$$V_t(\pi) = \mathbb{E}[\pi(t \mid X)\mu_t^\rho(X)].$$

*Then*

$$\mathrm{MSE}(\widehat{V}_{n,t}^{RR\text{-}IPW}(\pi)) - \mathrm{MSE}(\widehat{V}_{n,t}^{IPW}(\pi)) = b_t^2 - \frac{1}{n}\Delta_t,$$

*where*

$$b_t = \mathbb{E}\left[\pi(t \mid X)\left(m_t(X) - \mu_t^\rho(X)\right)\right]$$

*and*

$$\Delta_t = \mathbb{E}\left[\pi(t \mid X)^2\left(\frac{\sigma_{\rho,t}^2(X)}{e_t(X)} - \left(\frac{1}{e_t(X)} - 1\right)\{m_t(X)^2 - (\mu_t^\rho(X))^2\}\right)\right]$$
$$+ \mathrm{Var}(\pi(t \mid X)\mu_t^\rho(X)) - \mathrm{Var}(\pi(t \mid X)m_t(X)).$$

*Consequently, $\hat{V}_{n,t}^{RR-IPW}(\pi)$ has smaller mean-squared error than $\hat{V}_{n,t}^{IPW}(\pi)$ whenever*

$$b_t^2 < \frac{\Delta_t}{n}.$$

## C   Proofs

*Proof of Lemma 13.*

$$\mathbb{E}\left[\sum_t \mathbb{E}\left[\pi(t \mid X)\rho^\top \hat{Z}_t\right]\right] - \sum_t \mathbb{E}\left[\pi(t \mid X)\rho^\top \mathbb{E}\left[Z(t) \mid X\right]\right] \tag{17}$$

$$= \sum_t \mathbb{E}\left[\pi(t \mid X)\rho^\top \mathbb{E}\left[\hat{B}_t X \mid X\right]\right] - \mathbb{E}\left[\pi(t \mid X)\rho^\top \mathbb{E}\left[B_t X \mid X\right]\right] \tag{18}$$

$$= \sum_t \mathbb{E}\left[\pi(t \mid X)\rho^\top \mathbb{E}\left[\hat{B}_t X - B_t X \mid X\right]\right] \tag{19}$$

$$= \sum_t \mathbb{E}\left[\pi(t \mid X)\rho^\top \mathbb{E}\left[(\hat{B}_t - B_t)X \mid X\right]\right] \tag{20}$$

$$= 0 \tag{21}$$

where lines (14) and (15) are true by definition of $\hat{Z}_t$, and under Assumption 3. In the last line, we used the fixed-design unbiasedness of $\mathbb{E}[(\hat{B}_t - B_t)X \mid X] = 0$. Theorem 2.4 from Reinsel et al. (2022) gives the asymptotic normality result for $\hat{B}_t$ that implies asymptotic unbiasedness.

The result for $\hat{Y}$ holds using prediction error unbiasedness, where we have that $\mathbb{E}[(\hat{A}_t\hat{B}_t - A_t B_t)X \mid X] = 0$ by the asymptotic normality results of Equation 2.36 in Reinsel et al. (2022).

$\square$

*Proof of Lemma 14.*

$$\mathbb{E}\left[\sum_t \mathbb{E}\left[\pi(t \mid X)\frac{\mathbb{I}\left[T = t\right]}{e_t(X)}\rho^\top \hat{Z}_t\right]\right] - \sum_t \mathbb{E}\left[\pi(t \mid X)E\left[\frac{\mathbb{I}\left[T = t\right]}{e_t(X)}\rho^\top Z(t) \mid X\right]\right]$$

$$= \sum_t \mathbb{E}\left[\pi(t \mid X)\rho^\top \mathbb{E}\left[\frac{\mathbb{I}\left[T = t\right]}{e_t(X)}\hat{Z}_t \mid X\right]\right] - \sum_t \mathbb{E}\left[\pi(t \mid X)\frac{\mathbb{E}[\mathbb{I}\left[T = t\right] \mid X]}{e_t(X)}\rho^\top \mathbb{E}\left[Z(t) \mid X\right]\right] \tag{22}$$

$$= \sum_t \mathbb{E}\left[\pi(t \mid X)\rho^\top \frac{\mathbb{E}[\mathbb{I}\left[T = t\right] \mid X]}{e_t(X)}\mathbb{E}\left[\hat{B}_t X \mid X\right]\right] - \mathbb{E}\left[\pi(t \mid X)\rho^\top E\left[Z(t) \mid X\right]\right] \tag{23}$$

$$= \sum_t \mathbb{E}\left[\pi(t \mid X)\rho^\top \mathbb{E}\left[\hat{B}_t X \mid X\right]\right] - \mathbb{E}\left[\pi(t \mid X)\rho^\top E\left[B_t X \mid X\right]\right] \tag{24}$$

$$= \sum_t \mathbb{E}\left[\pi(t \mid X)\rho^\top \mathbb{E}\left[\hat{B}_t X - B_t X \mid X\right]\right]$$

$$= \sum_t \mathbb{E}\left[\pi(t \mid X)\rho^\top \mathbb{E}\left[(\hat{B}_t - B_t)X \mid X\right]\right]$$

$$= 0$$

where eq. (22) follows by iterated expectations, eq. (23) applies the pull-out property of conditional expectation, and eq. (24) is true by definition of the propensity score. The rest follows by the assumption on the data generating model Assumption 3. In the last line, we again used the fixed-design unbiasedness of $\mathbb{E}[(\hat{B}_t - B_t)X \mid X] = 0$. Theorem 2.4 from Reinsel et al. (2022) gives the asymptotic normality results for $\hat{B}_t$ that implies unbiasedness.

The result for $\hat{Y}$ holds using prediction error unbiasedness, where we have that $\mathbb{E}[(\hat{A}_t\hat{B}_t - A_t B_t)X \mid X] = 0$ by the asymptotic normality results of Equation 2.36 in Reinsel et al. (2022).

$\square$

*Proof of Corollary 5.* Let

$$\psi_t^{IPW}(\pi) = \pi(t \mid X) \frac{\mathbb{I}\left[T = t\right]}{e_t(X)} \rho^\top Y, \qquad \psi_t^{RR-IPW}(\pi) = \pi(t \mid X) \frac{\mathbb{I}\left[T = t\right]}{e_t(X)} \rho^\top \hat{\mu}_t^{RR}(X).$$

By iterated expectation,

$$\mathbb{E}[\psi_t^{IPW}(\pi)] = \mathbb{E}[\pi(t \mid X) \rho^\top \mu_t(X)], \qquad \mathbb{E}[\psi_t^{RR-IPW}(\pi)] = \mathbb{E}[\pi(t \mid X) \rho^\top \hat{\mu}_t^{RR}(X)].$$

Hence the bias of RR-IPW relative to IPW is

$$\mathbb{E}\left[\pi(t \mid X)\left(\rho^\top \hat{\mu}_t^{RR}(X) - \rho^\top \mu_t(X)\right)\right].$$

Since IPW is unbiased,

$$\mathrm{MSE}(\psi_t^{RR-IPW}(\pi)) - \mathrm{MSE}(\psi_t^{IPW}(\pi)) = \mathrm{Bias}(\psi_t^{RR-IPW}(\pi))^2 + \mathrm{Var}(\psi_t^{RR-IPW}(\pi)) - \mathrm{Var}(\psi_t^{IPW}(\pi)).$$

Now apply the law of total variance:

$$\mathrm{Var}(\psi_t^{IPW}(\pi)) - \mathrm{Var}(\psi_t^{RR-IPW}(\pi))$$
$$= \mathbb{E}\left[\mathrm{Var}(\psi_t^{IPW}(\pi) \mid X) - \mathrm{Var}(\psi_t^{RR-IPW}(\pi) \mid X)\right] + \mathrm{Var}\left(\pi(t \mid X) \rho^\top \mu_t(X)\right) - \mathrm{Var}\left(\pi(t \mid X) \rho^\top \hat{\mu}_t^{RR}(X)\right).$$

Finally,

$$\mathrm{Var}(\psi_t^{IPW}(\pi) \mid X) = \pi(t \mid X)^2 \left[\left(\frac{1}{e_t(X)} - 1\right)\left(\rho^\top \mu_t(X)\right)^2 + \frac{1}{e_t(X)} \sigma_{\rho,t}^2(X)\right],$$

and

$$\mathrm{Var}(\psi_t^{RR-IPW}(\pi) \mid X) = \pi(t \mid X)^2 \left(\frac{1}{e_t(X)} - 1\right)\left(\rho^\top \hat{\mu}_t^{RR}(X)\right)^2.$$

Substituting these expressions into the previous display gives the stated condition. ☐

*Proof of Corollary 5.* By Proposition 15, it suffices to upper bound

$$\left|\left(\rho^\top \hat{\mu}_t^{RR}(X)\right)^2 - \left(\rho^\top \mu_t(X)\right)^2\right|.$$

Using $a^2 - b^2 = (a - b)(a + b)$, we have

$$|a^2 - b^2| \leq |a - b|\,|a + b|.$$

Taking $a = \rho^\top \hat{\mu}_t^{RR}(X)$ and $b = \rho^\top \mu_t(X)$, the boundedness assumption implies

$$|a + b| \leq 2L_\rho,$$

and therefore

$$\left|\left(\rho^\top \hat{\mu}_t^{RR}(X)\right)^2 - \left(\rho^\top \mu_t(X)\right)^2\right| \leq 2L_\rho \left|\rho^\top \hat{\mu}_t^{RR}(X) - \rho^\top \mu_t(X)\right|.$$

Substituting this bound into Proposition 15 yields the claim. ☐

*Proof of Lemma 7.* The proof essentially follows by construction. Without loss of generality we omit the $\pi(t \mid X)$ term from the below. The first part of our estimator is the IPW estimator. Assuming the propensity scores are well specified, then we have that

$$\mathbb{E}\left[(1 - \tfrac{\mathbb{I}[T=1]}{e_1(X)}) \mid X\right] = 0.$$

This implies that $\forall h(X)$ functions,

$$\mathbb{E}\left[(1 - \tfrac{\mathbb{I}[T=1]}{e_1(X)})h(X)\right] = \mathbb{E}\left[\mathbb{E}[(1 - \tfrac{\mathbb{I}[T=1]}{e_1(X)}) \mid X]h(X)\right] = 0$$

The first equality is true by iterated expectation. This is mean-zero for all functions of X. We have

$$\mathbb{E}\left[\frac{\mathbb{I}[T=1]\,\rho^\top \hat{Z}}{e_1(X)} + \hat{D}_1 C_1 | X, T\right] = \underbrace{\mathbb{E}\left[\frac{\mathbb{I}[T=1]\,\rho^\top \hat{Z}}{e_1(X)}\right]}_{\text{consistent estimator}} + \underbrace{\mathbb{E}[\hat{D}_1 C_1]}_{\approx \text{ mean-zero noise}}$$

The proof of the result for $\mathbb{E}\left[\pi(t \mid X)\{\frac{\mathbb{I}[T=0]\rho^\top \hat{Z}}{e_0(X)} + \hat{D}_0 C_0\}\right]$ is similar. Thus our CV estimator $\hat{\phi}_t(\hat{D}_t)$ is unbiased.

The result for $\hat{Y}$ follows since the IPW estimator is also unbiased for $\hat{Y}$ by lemma 14. □

*Proof of Proposition 8.* We invoke standard assumptions in measurement noise of the dependent variable in OLS, which says that the measurement error is uncorrelated with $X$. Recalling that $\hat{B} = V^\top \Gamma^{1/2} \Sigma_{yx} \Sigma_{xx}^{-1}$, let $\epsilon_{\hat{Z}} = \hat{B}_t X - B_t X$ denote the measurement error in $\hat{Z}$; it is the estimation error from reduced rank regression. Then

$$\text{Cov}(\epsilon_{\hat{Z}}, X) = \mathbb{E}\left[\epsilon_{\hat{Z}} X\right] - \mathbb{E}\left[\epsilon_{\hat{Z}}\right] \mathbb{E}\left[X\right] = \mathbb{E}\left[\epsilon_{\hat{Z}} X\right]$$
$$= \mathbb{E}\left[\mathbb{E}\left[\epsilon_{\hat{Z}} \mid X\right] X\right] = 0$$

where the last inequality follows from $X$-conditional unbiased estimation of reduced rank regression parameters (Reinsel et al., 2022, p. 184).

Hence OLS with $\hat{Z}$ is consistent, in effect as if there were no measurement error (Wooldridge, 2001, 4.4.1).

□

*Proof of Theorem 9.* Assuming that $\mathbb{E}\left[C_t C_t^\top\right]$ is nonsingular, then $\hat{D}_t^* \in \arg\min_D \text{Var}\left[\phi(D)\right]$ is given by

$$\hat{D}_t^* = \mathbb{E}\left[C_t C_t^\top\right]^{-1} \mathbb{E}\left[\frac{\mathbb{I}[T=t]\,\rho^\top \hat{Z}_t}{e_t(X)}\right] \hat{B}_t X_t$$

and

$$\hat{D}_t = \mathbb{E}_n\left[C_t C_t^\top\right]^{-1} \mathbb{E}_n\left[\frac{\mathbb{I}[T=t]\,\rho^\top \hat{Z}_t}{e_t(X)}\right] \hat{B}_t X_t$$

We can invoke the results of Theorem 1 from (Glynn & Szechtman, 2002) and Proposition 8 to get asymptotic equivalence for our control variate estimator. □

*Proof of Proposition 15.* By iterated expectation,

$$\mathbb{E}[\psi_t^{IPW}(\pi) \mid X] = \pi(t \mid X)\rho^\top \mu_t(X), \qquad \mathbb{E}[\psi_t^{RR-IPW}(\pi) \mid X] = \pi(t \mid X)\rho^\top \hat{\mu}_t^{RR}(X),$$

which gives the bias expression.

For the variance, use $\text{Var}(W \mid X) = \mathbb{E}[W^2 \mid X] - \mathbb{E}[W \mid X]^2$. Since

$$\mathbb{E}[(\rho^\top Y)^2 \mid X, T = t] = (\rho^\top \mu_t(X))^2 + \sigma_{\rho,t}^2(X),$$

we obtain

$$\text{Var}(\psi_t^{IPW}(\pi) \mid X) = \pi(t \mid X)^2 \left[ \left( \frac{1}{e_t(X)} - 1 \right)(\rho^\top \mu_t(X))^2 + \frac{1}{e_t(X)} \sigma_{\rho,t}^2(X) \right].$$

Similarly, because $\rho^\top \hat{\mu}_t^{RR}(X)$ is fixed given $X$,

$$\text{Var}(\psi_t^{RR-IPW}(\pi) \mid X) = \pi(t \mid X)^2 \left( \frac{1}{e_t(X)} - 1 \right)(\rho^\top \hat{\mu}_t^{RR}(X))^2.$$

Subtracting yields the displayed variance difference. The final claim follows from

$$\text{MSE} = \text{Bias}^2 + \text{Var}.$$

$\square$

# D   Generalization bounds for policy learning

## D.1   Preliminaries

First we begin by collecting some technical results used without proof.

**Lemma 16** (Proposition 15 from Bunea et al. (2011))**.** *Let $E$ be a $n \times k$ matrix with independent subGaussian entries $E_{ij}$ with subGaussian moment $\Gamma_E$. Let $X$ be an $n \times p$ matrix of rank $q$ and let $P = X(X^\top X)^- X^\top$ be the projection matrix on $R[X]$. Then, for each $x > 0$, and large enough $C_0$, we have*

$$\Pr(d_1^2(PE) \leq C_0(k + q)) \leq 2 \exp\left(-(k + q)\right),$$

*where $d_1(PE)$ is the largest singular value of the projected noise matrix $PE$.*

**Lemma 17** (Error bounds for the RRR rank selection criterion estimator (Bunea et al. (2011), Theorem 5))**.** *This bound is derived for the fit*

$$\|\hat{A}_t \hat{B}_t X - A_t B_t X\|_F^2 := \sum_i \sum_j \left\{ (\hat{A}_t \hat{B}_t X)_{ij} - (A_t B_t X)_{ij} \right\}^2$$

*based on the restricted rank estimator for each value of the rank $r$. Then for large enough constants $C_1, C_2$ and with probability one, we have*

$$\|\hat{A}_t^{(r)} \hat{B}_t^{(r)} X - A_t B_t X\|_F^2 \leq C_1 \sum_{j > r} d_j^2(A_t B_t X) + C_2 r d_1^2(PE),$$

*where $\sum_{j > r} d_j^2(A_t B_t X)$ is an approximation error and $r d_1^2(PE)$ is a stochastic term that concentrates the error, is increasing in $r$, and can be bounded by a constant times $r(k + q)$ by lemma 16.*

This bound holds when we choose the rank based on the rank selection criterion described in Section 2 of Bunea et al. (2011). Although we did not choose our rank in this way, we find that empirically the choice of rank does not matter much in our experiments.

**Lemma 18** (Bernstein's inequality)**.** *Suppose $X_1, \ldots, X_n$ are i.i.d. with 0 mean, variance $\sigma^2$ and $\|X_i\| \leq M$ almost surely. Then with probability at least $1 - \delta$, we have*

$$\left| \frac{1}{n} \sum_{i=1}^n X_i \right| \leq \sqrt{\frac{2\sigma^2}{n} \log\left(\frac{2}{\delta}\right)} + \frac{2M}{3n} \log\left(\frac{2}{\delta}\right).$$

### D.2    Results

**Assumption 19** (Bounded outcomes). *For any $X, T$, the outcomes $|Y_i| \leq L$ almost surely.*

**Assumption 20** (Overlap in propensity scores). *For any $X, T$, $0 < e_t(X) < 1$.*

**Theorem 21** (Deviation bound for CV). *Under Assumption 19 and Assumption 20, further assume that $\left|\hat{V}_{\hat{Y}}^{RR-CV}(\pi)\right| \leq L \cdot 1$. Assume that either the propensity scores are known or traditionally unbiased conditional on X. Then with probability at least $1 - \delta$, we have*

$$\left|\hat{V}_{\hat{Y}}^{RR-CV}(\pi) - V_Y(\pi)\right| \leq \sqrt{2\log\left(\frac{2}{\delta}\right) \cdot \frac{\mathrm{Var}[\phi^Y(D^*;\pi)]}{n}} + \frac{2L}{3n}\log\left(\frac{2}{\delta}\right) + \sqrt{\frac{2}{n}}\{C_1\sum_{j>r}d_j^2(A_tB_tX) + C_2rd_1^2(PE)\}^{\frac{1}{2}}.$$

*and*

$$\left|\hat{V}_{\hat{Z}}^{RR-CV}(\pi) - V_Z(\pi)\right| \leq \sqrt{2\log\left(\frac{2}{\delta}\right) \cdot \frac{\mathrm{Var}[\phi^Z(D^*;\pi)]}{n}} + \frac{2L}{3n}\log\left(\frac{2}{\delta}\right) + \sqrt{\frac{2}{n}}\{C_1\sum_{j>r}d_j^2(B_tX) + C_2rd_1^2(PE)\}^{\frac{1}{2}}.$$

**Corollary 22** (Deviate bound for CV with estimated propensity scores). *Under the conditions of theorem 21, suppose that the propensity scores are estimated and satisfy: (i) $\|\hat{e}_t - e_t\|_\infty \leq \epsilon_n$ with probability $1 - \delta$, and (ii) they are bounded away from 0 and 1 where $\epsilon \leq \hat{e}_t(X) \leq 1 - \epsilon$ almost surely for some $\epsilon > 0$. Then with probability at least $1 - 2\delta$, we have*

$$\left|\hat{V}_{\hat{Y}}^{RR-CV}(\pi) - V_Y(\pi)\right| \leq \sqrt{2\log\left(\frac{2}{\delta}\right) \cdot \frac{\mathrm{Var}[\phi^Y(D^*;\pi)]}{n}} + \frac{2L}{3n}\log\left(\frac{2}{\delta}\right) + \sqrt{\frac{2}{n\epsilon^2}}\{C_1\sum_{j>r}d_j^2(A_tB_tX) + C_2rd_1^2(PE)\}^{\frac{1}{2}} + O\left(\frac{\epsilon_n}{\epsilon}\right).$$

*and*

$$\left|\hat{V}_{\hat{Z}}^{RR-CV}(\pi) - V_Z(\pi)\right| \leq \sqrt{2\log\left(\frac{2}{\delta}\right) \cdot \frac{\mathrm{Var}[\phi^Z(D^*;\pi)]}{n}} + \frac{2L}{3n}\log\left(\frac{2}{\delta}\right) + \sqrt{\frac{2}{n\epsilon^2}}\{C_1\sum_{j>r}d_j^2(B_tX) + C_2rd_1^2(PE)\}^{\frac{1}{2}} + O\left(\frac{\epsilon_n}{\epsilon}\right).$$

**Remark 1** (Variance reduction for regression control variates). *The bound above is dependent on the worst-case variance across the policy class. By construction, our regression control variate estimator is one that minimizes variance. To see this, note that the variance of the estimator with control variates is:*

$$\mathrm{Var}[\phi^{CV}(D^*)] = \mathrm{Var}\left[\frac{\mathbb{I}[T=1]\rho^\top \hat{Z}}{e_1(X)}\right] - 2D^*\mathbb{E}\left[\frac{\mathbb{I}[T=1]\rho^\top \hat{Z}}{e_1(X)}\right]C + D^\top\mathbb{E}[CC^\top]D.$$

*Note that $D = 0$ is a feasible solution for this optimization problem but it is not optimal, therefore our choice of $D^*$ has a lower variance than the IPW estimator.*

$$\mathrm{Var}[\phi^{CV}(D^*)] \leq \mathrm{Var}\left[\frac{I[T=t]\rho^T\hat{Z}}{e_t(X)}\right]$$

*The result holds for $\hat{Y}$ as the proof relies only on our choice of $D^*$.*

**Corollary 23** (Uniform deviation bound for policy optimization). *Given a finite policy class $\Pi$ with $|\Pi| = N$, assume that the conditions of Theorem 21 are satisfied for each $\pi \in \Pi$. Then with probability at least $1 - \delta$, we have*

$$\sup_{\pi \in \Pi}\left|\hat{V}_{\hat{Y}}^{RR-CV}(\pi) - V_Y(\pi)\right| \leq \sqrt{2\log\left(\frac{2N}{\delta}\right) \cdot \frac{\mathrm{Var}[\phi^Y(D^*;\pi)]}{n}} + \frac{2L}{3n}\log\left(\frac{2N}{\delta}\right) + \sqrt{\frac{2}{n}}\{C_1\sum_{j>r}d_j^2(A_tB_tX) + C_2rd_1^2(PE)\}^{\frac{1}{2}}.$$

**Theorem 24** (Generalization error). *With probability $1 - \delta$, we can bound our estimate of $V^{RR-CV}(\hat{\pi}_n)$ by*

$$V^{RR-CV}(\hat{\pi}_n; \hat{Y}) \geq V_Y(\pi^*) - \sqrt{2\log\left(\frac{2N}{\delta}\right) \cdot \frac{\sup_{\pi \in \Pi}\{\mathrm{Var}[\phi^Y(D^*; \pi)]\}}{n}} + \frac{2L}{3n}\log\left(\frac{2N}{\delta}\right) + \sqrt{\frac{2}{n}}\{C_1 \sum_{j>r} d_j^2(A_t B_t X) + C_2 r d_1^2(PE)\}$$

### D.3 Proofs

*Proof of Theorem 21.* To analyze the finite-sample error, we decompose by $\pm\mathbb{E}[\hat{V}_{\hat{Y}}^{RR-CV}(\pi)]$ to get two terms that represent the generalization error and estimation error,

$$\left|\hat{V}_{\hat{Y}}^{RR-CV}(\pi) - V_Y(\pi)\right| \leq \left|\hat{V}_{\hat{Y}}^{RR-CV}(\pi) - V_Y(\pi) \pm \mathbb{E}[\hat{V}_{\hat{Y}}^{RR-CV}(\pi)]\right|$$

$$\leq \underbrace{\left|\hat{V}_{\hat{Y}}^{RR-CV}(\pi) - \mathbb{E}[\hat{V}_{\hat{Y}}^{RR-CV}(\pi)]\right|}_{\text{generalization error}} + \underbrace{\left|\mathbb{E}[\hat{V}_{\hat{Y}}^{RR-CV}(\pi)] - V_Y(\pi)\right|}_{\text{estimation error}}$$

We start by bounding the first term. By Lemma 7, we know that $\hat{V}_{\hat{Y}}^{RR-CV}(\pi)$ is a sum of $n$ i.i.d. random variables, each with mean $\mathbb{E}[\hat{V}_{\hat{Y}}^{RR-CV}(\pi)]$. We also have bounds on both the range and variance of the CV estimator from Lemma 7 and Remark 1. We can immediately obtain an upper bound by the application of Bernstein's inequality in Lemma 18 uniformly over the policy space

$$\left|\hat{V}_{\hat{Y}}^{RR-CV}(\pi) - \mathbb{E}[\hat{V}_{\hat{Y}}^{RR-CV}(\pi)]\right| \leq \sqrt{2\log\left(\frac{2}{\delta}\right) \cdot \frac{\mathrm{Var}[\phi^Y(D^*; \pi)]}{n}} + \frac{2L}{3n}\log\left(\frac{2}{\delta}\right)$$

Next, we bound the second term as follows

$$\left|V_Y(\pi) - \mathbb{E}[\hat{V}_{\hat{Y}}^{RR-CV}(\pi)]\right| = \left|\sum_t \mathbb{E}\left[\pi(t \mid X)\frac{\mathbb{I}[T=t]}{e_t(X)}\rho^\top Y | X, T\right] - \sum_t \mathbb{E}\left[\pi(t \mid X)\frac{\mathbb{I}[T=t]}{e_t(X)}\rho^\top \hat{Y} | X, T\right]\right|$$

$$= \left|\mathbb{E}\left[\sum_t \pi(t \mid X)(\rho^\top A_t B_t X - \rho^\top \hat{A}_t \hat{B}_t X)\right]\right|$$

$$= \left|\mathbb{E}\left[\pi(0 \mid X)\rho^\top (A_0 B_0 - \hat{A}_0 \hat{B}_0)X + \pi(1 \mid X)\rho^\top B_1 - \hat{A}_1 \hat{B}_1)X\right]\right|$$

$$\leq \left|\mathbb{E}[\pi(0 \mid X)\rho^\top (A_0 B_0 - \hat{A}_0 \hat{B}_0)X]\right| + \left|\mathbb{E}[\pi(1 \mid X)\rho^\top (A_1 B_1 - \hat{A}_1 \hat{B}_1)X]\right|$$

By the Cauchy-Schwarz inequality, we relate these terms above to the prediction error bounds in lemma 17, where

$$\left|\mathbb{E}\left[\pi(t \mid X)\rho^\top (A_t B_t - \hat{A}_t \hat{B}_t)X\right]\right|^2 \leq \underbrace{\mathbb{E}[\pi(t \mid X)^2]}_{<1}\underbrace{\mathbb{E}[((A_t B_t - \hat{A}_t \hat{B}_t)X\rho)^2]}_{\leq \frac{1}{n}\|((A_t B_t - \hat{A}_t \hat{B}_t)X\rho)\|_2^2} \tag{25}$$

Then by lemma 17, we can bound this term as

$$\leq \frac{2}{n}\{C_1 \sum_{j>r} d_j^2(A_t B_t X) + C_2 r d_1^2(PE)\}$$

Combining the two terms, we get the final bound

$$\left|\hat{V}_{\hat{Y}}^{RR-CV}(\pi) - V_Y(\pi)\right| \leq \sqrt{2\log\left(\frac{2}{\delta}\right) \cdot \frac{\mathrm{Var}[\phi^Y(D^*; \pi)]}{n}} + \frac{2L}{3n}\log\left(\frac{2}{\delta}\right) + \sqrt{\frac{2}{n}}\{C_1 \sum_{j>r} d_j^2(A_t B_t X) + C_2 r d_1^2(PE)\}^{\frac{1}{2}}$$

$$\square$$

*Proof of Corollary 22.* Starting with the bound on the first term, the variance of CV estimator with estimated propensities is biased and thus the variances term is bounded as follows: We can decompose the CV estimator into two terms, the first is a consistent estimator and then the bias term as follows

$$\mathbb{E}\left[\frac{\mathbb{I}\left[T=t\right]\rho^\top \hat{Z}}{\hat{e}_t(X)} + \hat{D}_t C_t\right] = \mathbb{E}\left[\frac{\mathbb{I}\left[T=t\right]\rho^\top \hat{Z}}{\hat{e}_t(X)} + \left(\frac{\hat{e}_t(X) - \mathbb{I}\left[T=t\right]}{\hat{e}_t(X)}\right)(\hat{B}_t^\top X^\top X \hat{B}_t)^{-1}\hat{B}_t^\top X^\top X \hat{B}_t\right]$$

$$= \underbrace{\mathbb{E}\left[\frac{\mathbb{I}\left[T=t\right]\rho^\top \hat{Z}}{\hat{e}_t(X)}\right]}_{\text{consistent estimator}} + \underbrace{\mathbb{E}\left[\left(\frac{\hat{e}_t(X) - \mathbb{I}\left[T=t\right]}{\hat{e}_t(X)}\right)\rho^\top I\right]}_{\text{bias term}=O(\frac{\epsilon_n}{\epsilon})}$$

Note that the consistent estimator is the IPW estimator with estimated propensities and it is easy to see that this remains unbiased since $\mathbb{E}[(\hat{B}_t - B_t)X|X] = 0$. Additionally, we see that the variance of the CV estimator with estimated propensities remains lower than the variance of the IPW estimator

$$\text{Var}[\phi^{CV}(\hat{D}^*;\pi)] \leq \text{Var}\left[\frac{\mathbb{I}\left[T=t\right]\rho^\top \hat{Z}}{\hat{e}_t(X)}\right] \leq \frac{1}{\epsilon^2}\text{Var}[\mathbb{I}\left[T=t\right]\rho^\top \hat{Z}],$$

where the second inequality follows by $\hat{e}_t(X) \geq \epsilon$. Combined, we can apply Bernstein's inequality to the mean zero, consistent estimator term and then add the bias term to the final bound. For the bound on the second term, following a similar form to Equation (25), we get

$$\left|\mathbb{E}\left[\pi(t \mid X)\frac{\mathbb{I}\left[T=t\right]}{\hat{e}_t(X)}\rho^\top(A_t B_t - \hat{A}_t\hat{B}_t)X\right]\right|^2 \leq \underbrace{\mathbb{E}[\pi(t \mid X)^2]}_{<1}\underbrace{\mathbb{E}[\left(\frac{\mathbb{I}\left[T=t\right]}{\hat{e}_t(X)}\right)^2]}_{\leq \frac{1}{\epsilon^2}}\underbrace{\mathbb{E}[((A_t B_t - \hat{A}_t\hat{B}_t)X\rho)^2]}_{\leq \frac{1}{n}\|((A_t B_t - \hat{A}_t\hat{B}_t)X\rho)\|_2^2}$$

Then by lemma 17, we can bound this term as

$$\leq \frac{2}{n\epsilon^2}\{C_1 \sum_{j>r} d_j^2(A_t B_t X) + C_2 r d_1^2(PE)\}$$

Combining the two terms, we get the final bound is

$$\left|\hat{V}_{\hat{Y}}^{RR-CV}(\pi) - V_Y(\pi)\right| \leq \sqrt{2\log\left(\frac{2}{\delta}\right) \cdot \frac{\text{Var}[\phi^Y(D^*;\pi)]}{n}} + \frac{2L}{3n}\log\left(\frac{2}{\delta}\right) + \sqrt{\frac{2}{n\epsilon^2}}\{C_1 \sum_{j>r} d_j^2(A_t B_t X) + C_2 r d_1^2(PE)\}^{\frac{1}{2}} + O(\frac{\epsilon_n}{\epsilon}).$$

$$\square$$

*Proof of Theorem 24.* We can define $\hat{\pi}_n = \arg\min_{\pi \in \Pi} \hat{V}_{CV}(\pi;\hat{Y})$. Applying Corollary 23 twice, once with $\hat{\pi}_n$ and once with $\pi^*$, and using a uniform upper bound on the variance for all policies in $\Pi$, we obtain

$$V_Y(\hat{\pi}_n) \geq V_Y(\pi^*) - \sqrt{2\log\left(\frac{2N}{\delta}\right) \cdot \frac{\sup_{\pi \in \Pi}\{\text{Var}[\phi^Y(D^*;\pi)]\}}{n}} + \frac{2L}{3n}\log\left(\frac{2N}{\delta}\right) + \sqrt{\frac{2}{n}}\{C_1 \sum_{j>r} d_j^2(A_t B_t X) + C_2 r d_1^2(PE)\}^{\frac{1}{2}}$$

$$\square$$

# E  Additional Experiments and Results

## E.1  Synthetic Data

Figure 6 compares the variances as the noise level of the observed outcomes $Y$ increases. Figure 7 compares the variances as the dimensions of $Y$ and $Z$, where the dimensions of $Y$ are always greater than $Z$. In

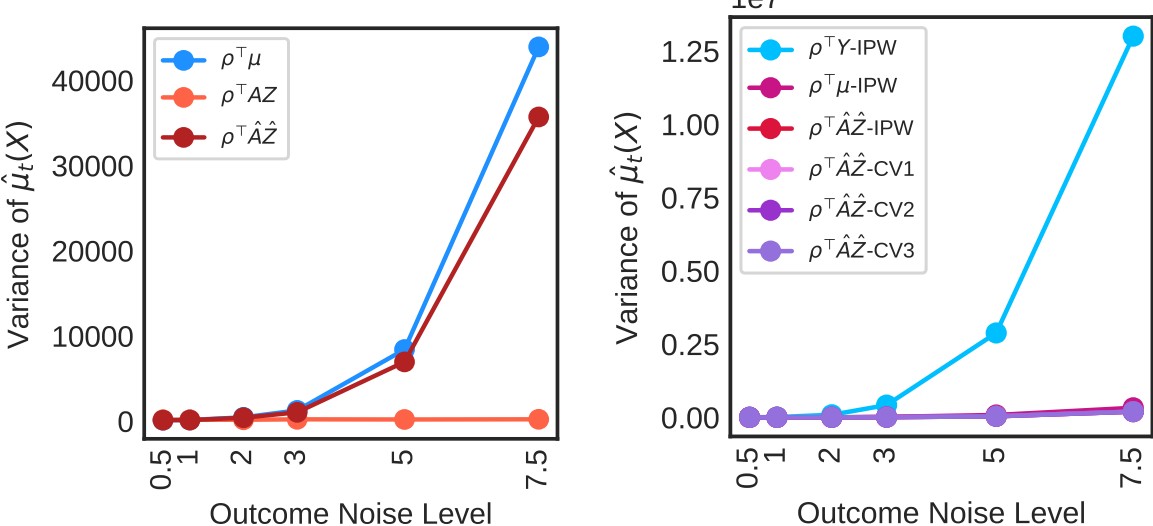

Figure 6: **Left:** Comparison of variances averaged over 100 datasets as the noise in $Y$ increases using the direct estimators for $\rho Y$. **Right:** Comparison of variances as the noise level in $Y$ increases for the IPW and control variate estimators. *Lower is better.*

Figure 7, we explore how the variance improvements change in response to varying dimensions of underlying $Z$ and observed $Y$.

In Figure 3, we focus on comparing MSE for out-of-sample policy value of policies that are optimal using a scalarized standard doubly-robust method, our control variate with the denoised estimator, and the naive Direct Method (without reduced rank regression). Our method improves upon these other doubly-robust or ablated control variate baselines, especially at small sample sizes.

Finally, we include a few details about other possible vectors of control variates. (We didn't find any significant diferences among these). Instead of chooding $h_t(X) = \hat{B}_t X$, we could also choose the following:

1. $h_t(X) = \mathbb{E}\left[Y\right]$, no regression control variate

2. $h_t(X) = \mathbb{E}\left[Y \mid X\right]$, marginalizing over $T$ in the observational data, with regression control variates

3. $h_t(X) = \rho^\top \mathbb{E}\left[\hat{Z} \mid X\right]$, marginalizing over $T$ in the observational data, with regression control variates

4. $h_t(X) = \mathbb{E}\left[Y \mid T, X\right]$, with regression control variates

### E.2 Sahel Adaptive Social Protection Program Multi-Country RCT Dataset

The study is described in Bossuroy et al. (2022a)[4]. We provide additional exploratory analysis for finding heterogeneous treatment effects in the dataset. We complete this analysis using the EconML package hosted by Microsoft.

#### E.2.1 Data cleaning and preprocessing

**Features** The dataset includes different views on both baseline outcomes and background demographic data, which was collected for a subset of households. We merged the two data sources and kept the households with full information across both baseline and background features. Additionally, after some exploratory

---

[4]Data available for download at `https://microdata.worldbank.org/index.php/catalog/4294/get-microdata`

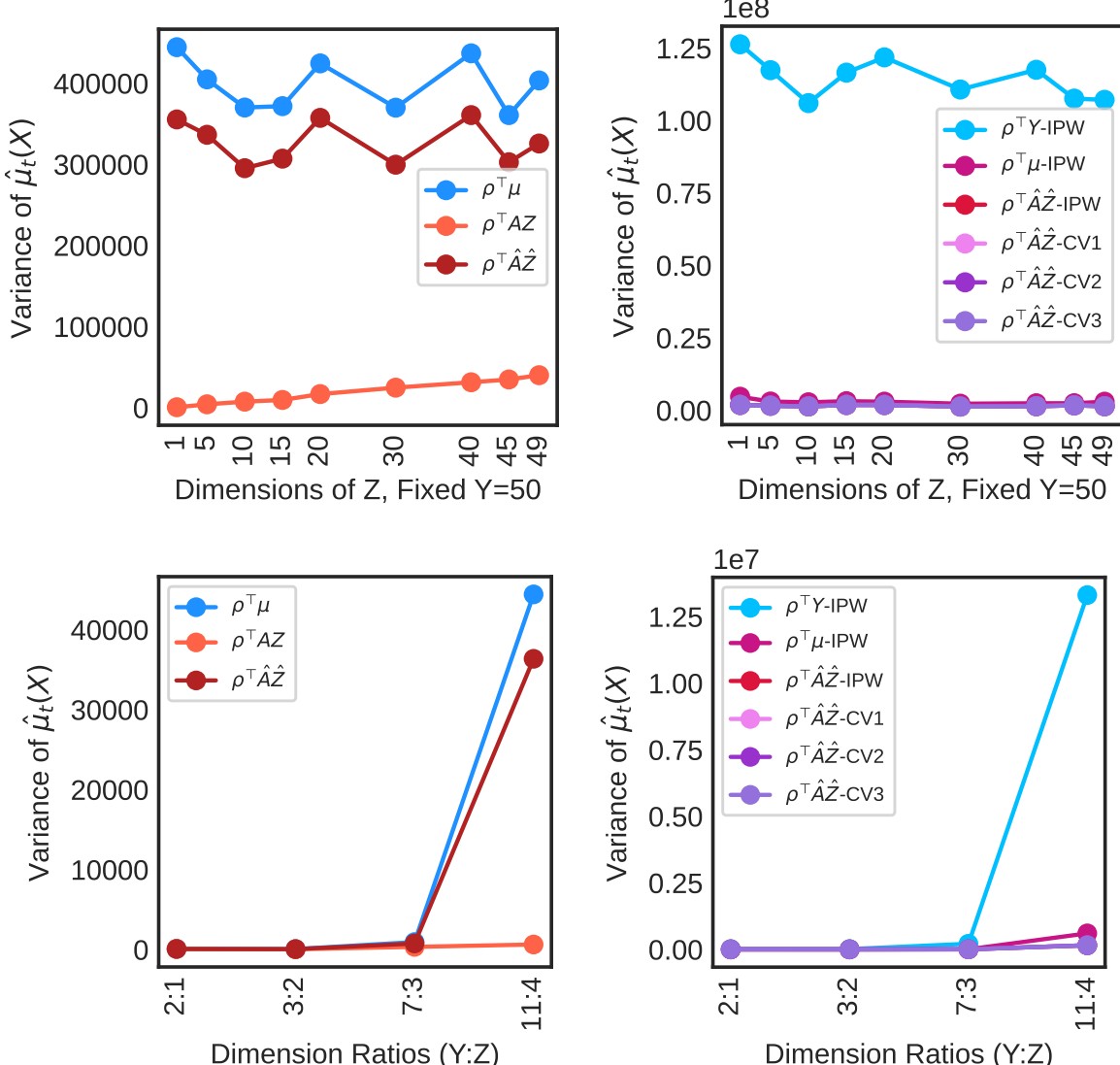

Figure 7: **First Left:** Comparison of variances averaged over 100 datasets as the dimensions of $Y$ and $Z$ increase using the direct estimators for $\rho Y$. The ratio of dimensions is $d_Y/d_Z$ where the dimensions of $Y$ are always greater than $Z$. **Second Left:** Comparison of variances as the ratio of dimensions of $Y$ to $Z$ is increasing for the IPW and control variate estimators. **First Right:** Comparison of variances averaged over 100 datasets as the dimensions of $Z$ increase and the dimension of $Y$ remains fixed at $k = 50$ for $\rho Y$. **Second Right:** Comparison of variances as the dimensions of $Z$ increase and the dimensions of $Y$ remain fixed for the IPW and control variate estimators. *Lower is better.*

analysis, we find that the background demographic data is more predictive, so we narrow down the feature space even further based on availability of both sets of features.

**Treatments**   Originally, this study had four treatment arms to address the financial and psycho-social dimensions of poverty. The study previously identified 6 productive interventions that would be administered through a countries national cash transfer program. They are (1) needs-based individualized coaching and group-based facilitation (2) community sensitization on aspirations and social norms (3) facilitation of community savings groups (4) micro-entrepreneurship training (5) behavioral skills training (6) one-time

Table 4: Sahel dataset feature list and feature descriptions.

| Feature Name | Description |
| --- | --- |
| less_depressed_bl | Depression Baseline: 4 questions from (CESD-R-10), (0-7, recode to 1-4);sum, reversed |
| less_disability_bl | Disability Baseline: 4 questions from the SRQ-20 (Self-report questionnaire), (recoded from [1-4] to 0/1), (neurotic, stress-related disorders), reversed |
| stair_satis_today_bl | Life Satisfaction: Cantril's latter of life satisfaction (1-10) |
| ment_hlth_index_trim_bl | One item mental health assessment: Productive beneficiary mental health self-assessment (1-5), standardized |
| hhh_fem | Household head is female |
| pben_fem | Productive beneficiary is female |
| hhh_age | Household head age |
| pben_age | Productive beneficiary age |
| hhh_poly | Household head is in polygamous relationship (male or female) |
| pben_poly | Productive beneficiary is in polygamous relationship (male or female) |
| pben_handicap | Beneficiary has handicap |
| phy_lift_hhh | How difficult is it for the household head to lift a 10 kg bag? |
| phy_walk_hhh | How difficult is it for the household head to walk 4 hours without resting? |
| phy_work_hhh | How difficult is it for the household head to work in the fields all day? |
| phy_lift_pben | How difficult is it for the beneficiary to lift a 10 kg bag? |
| phy_walk_pben | How difficult is it for the beneficiary to walk 4 hours without resting? |
| phy_work_pben | How difficult is it for the beneficiary to work in the fields all day? |
| pben_edu | Productive beneficiary years of education |
| pben_prim | Productive beneficiary completed primary school |
| pben_lit | Productive beneficiary is literate |
| hhh_edu | Household head years of education |
| hhh_prim | Household head completed primary school |
| hhh_lit | Household head is literate |
| hou_room | Number of rooms in house |
| hou_hea_min | Minutes needed to get to nearest health center (usual mode of transport)? |
| hou_mar_min | Minutes needed to get to nearest market (usual mode of transport)? |
| hou_wat_min | Minutes needed to get to main source of drinking water (usual mode of transport) |
| km_to_com | Distance to capital of commune (km) |

cash grant. These interventions were groups into the four treatments arms for impact evaluation. All arms received the regular cash transfer program. All treatment arms $(T_c, T_s, T_f)$, not including the control arm, received the core package (savings groups, business training and coaching) which corresponds to interventions (1), (3), and (4). The next treatment arm receives all of the above and the social package $T_s$ which includes life skills training and community sensitization or interventions (2) and (5) respectively. Treatment arm $T_c$ receives all of the above, but instead of the social package, they receive the capital package which includes micro-entrepreneurship training and a cash grant or interventions (4) and (6) respectively. The final treatment arm $T_f$ receives all of the interventions, which includes the core package, the social package and the capital package. For our experiments, we focus on the binary treatment between the control group and those that received the full treatment arm $T_f$.

### E.2.2 Finding Heterogeneous Treatment Effects

We conduct an exploratory analysis on the seven outcomes of interest to confirm the existence of heterogeneous treatment effects in the Sahel dataset. We provide descriptive information about the estimated causal effect using causal random forests (Athey & Wager, 2020) and OLS regression with interaction terms between the treatment and covariates. The regression results are summarized in Table 5 where we find a few significant interaction terms between the treatment and the features. This is an indication of heterogeneity across the seven outcomes. In Figure 8 - Figure 14, the first plot shows the estimates of the causal effect and the second plot shows the distribution of the outcome in the test set.

**Data transformations for reduced rank regression** We log-transform the outcomes: beneficiary wages, beneficiary productive revenue, business asset value, investments, and savings, as is common in econometrics: histograms in the appendix illustrate the right-tailed distribution. Lastly, we tune the number of factors that we estimate for $\hat{Y}$, though this ends up quite similar to our results in the main paper with two factors.

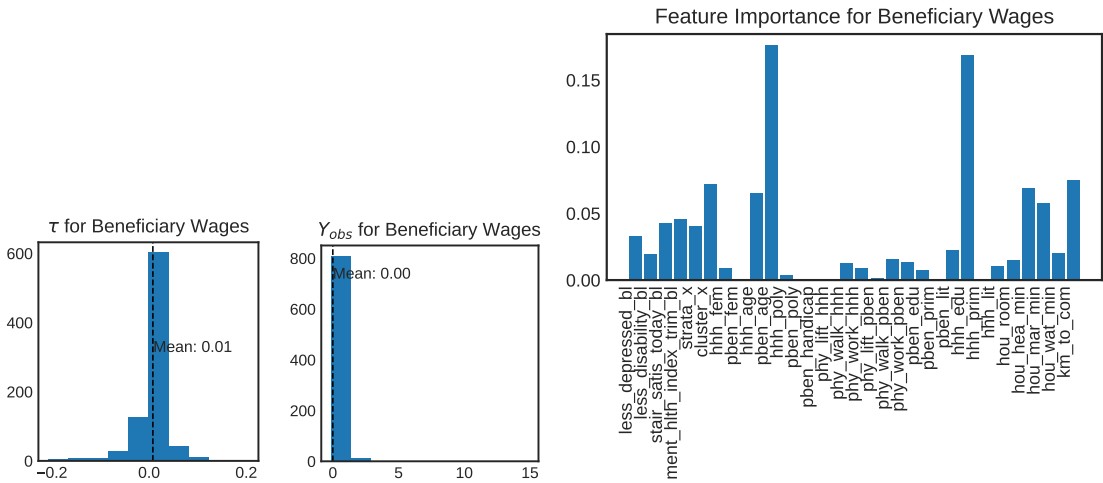

Figure 8: Distribution of treatment effect for beneficiary wages, distribution of observed outcome data in the test set, plot of feature importance from causal forest on training set

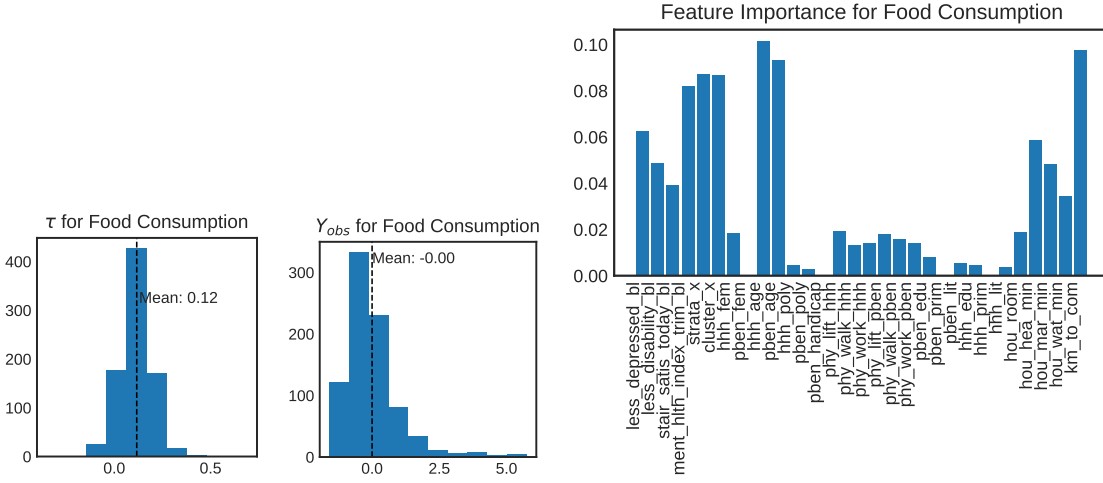

Figure 9: Distribution of treatment effect for food consumption, distribution of observed outcome data in the test set, plot of feature importance from causal forest on training set

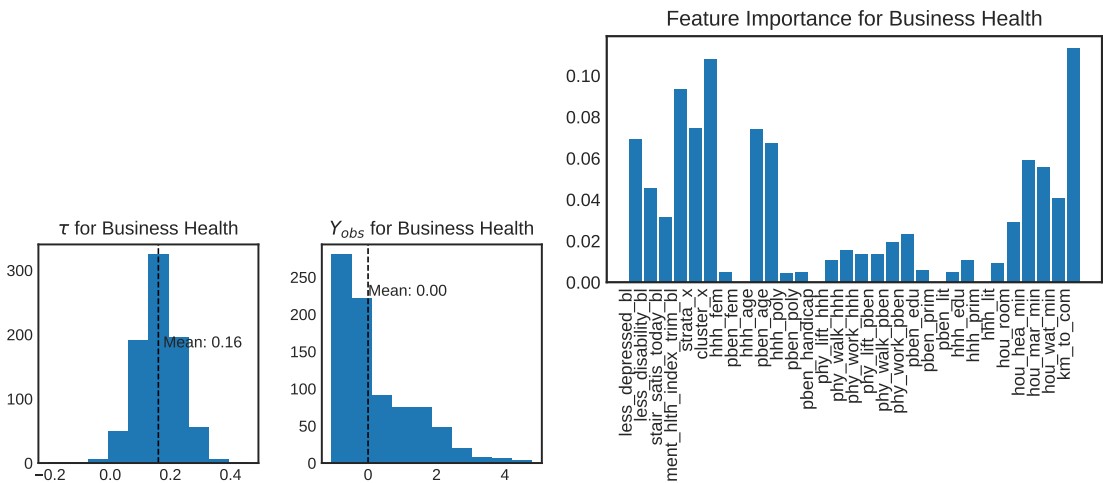

Figure 10: Distribution of treatment effect for business health, distribution of observed outcome data in the test set, plot of feature importance from causal forest on training set

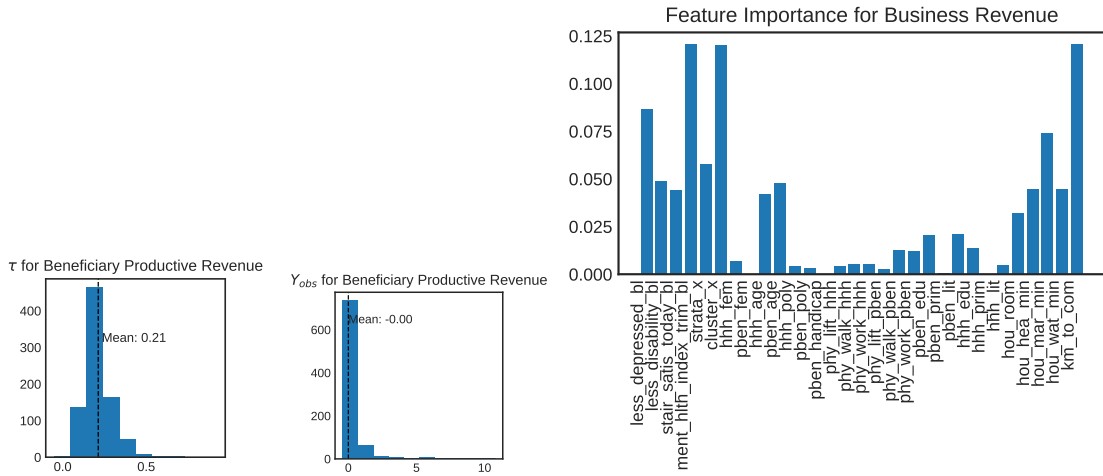

Figure 11: Distribution of treatment effect for business revenue, distribution of observed outcome data in the test set, plot of feature importance from causal forest on training set

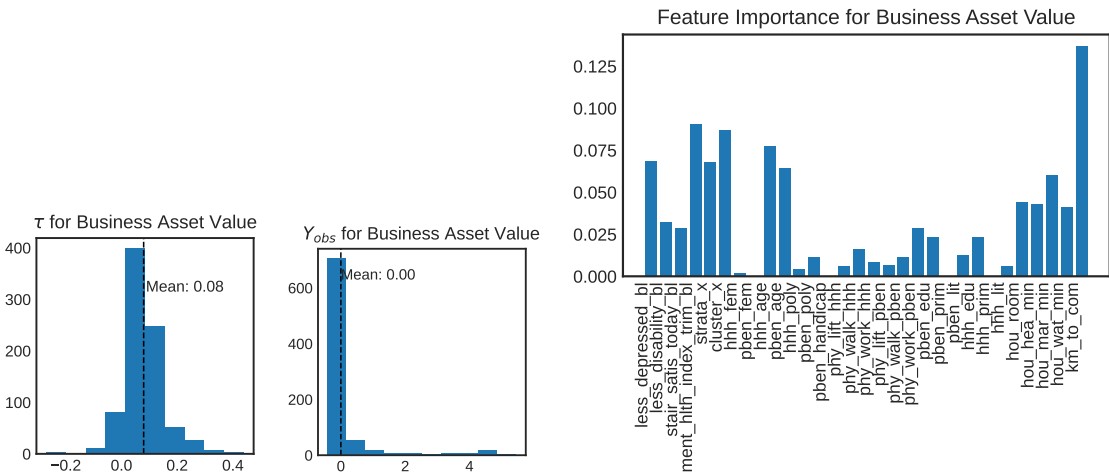

Figure 12: Distribution of treatment effect for business asset value, distribution of observed outcome data in the test set, plot of feature importance from causal forest on training set

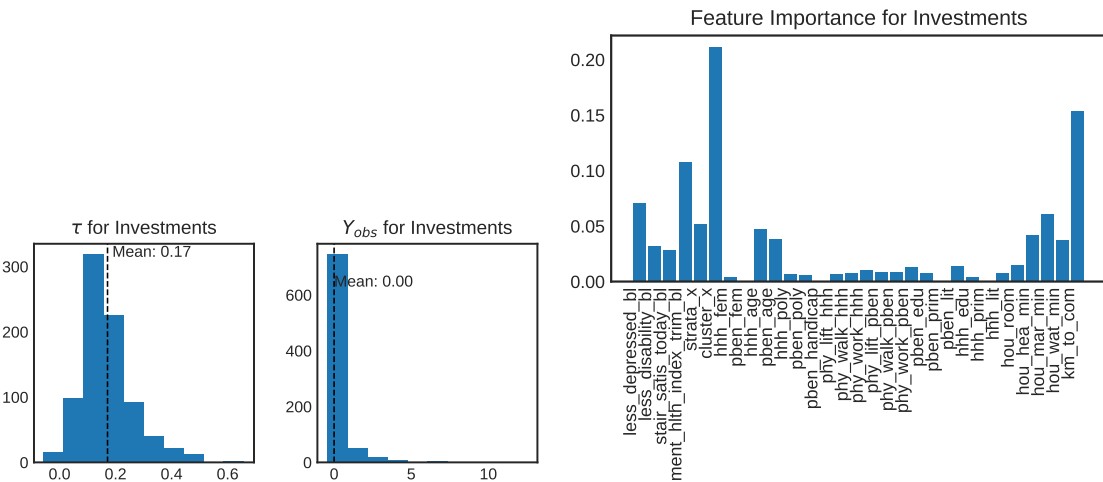

Figure 13: Distribution of treatment effect for investments, distribution of observed outcome data in the test set, plot of feature importance from causal forest on training set

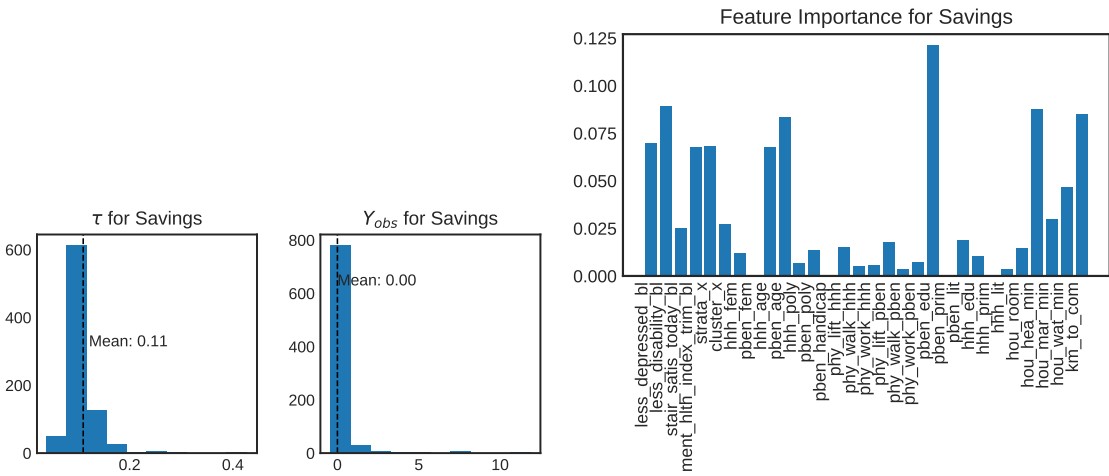

Figure 14: Distribution of treatment effect for savings, distribution of observed outcome data in the test set, plot of feature importance from causal forest on training set

Table 5: OLS regression table output for each scaled outcome, treatment, and interaction terms with a subset of relevant covariates. The standard errors in parentheses. * $p < .1$, ** $p < .05$, ***$p < .01$

| | Beneficiary Wages | Food Consumption | Business Health | Beneficiary Productive Revenue | Business Asset Value | Investments | Savings |
|---|---|---|---|---|---|---|---|
| Intercept | 0.3624* | 0.1321 | -0.4559** | 0.3673* | -0.0356 | 0.1010 | -0.3024 |
| | (0.2131) | (0.2120) | (0.2148) | (0.2064) | (0.2118) | (0.2071) | (0.2103) |
| treatment | 0.1178 | -0.1631 | 0.1452 | 0.2315 | 0.1015 | -0.2885 | 0.2275 |
| | (0.3163) | (0.3121) | (0.3174) | (0.3063) | (0.3146) | (0.3075) | (0.3122) |
| hhh_age | -0.0017 | -0.0003 | 0.0031 | 0.0001 | 0.0037** | 0.0016 | 0.0032* |
| | (0.0019) | (0.0018) | (0.0019) | (0.0018) | (0.0018) | (0.0018) | (0.0018) |
| treatment:hhh_age | 0.0038 | 0.0019 | -0.0004 | 0.0007 | -0.0025 | -0.0023 | -0.0036 |
| | (0.0026) | (0.0026) | (0.0027) | (0.0025) | (0.0026) | (0.0026) | (0.0026) |
| pben_age | -0.0012 | 0.0018 | 0.0048** | 0.0027 | -0.0040** | 0.0006 | -0.0006 |
| | (0.0019) | (0.0019) | (0.0022) | (0.0019) | (0.0019) | (0.0019) | (0.0019) |
| treatment:pben_age | -0.0073*** | -0.0047* | -0.0059** | -0.0009 | 0.0041 | 0.0019 | 0.0015 |
| | (0.0027) | (0.0027) | (0.0029) | (0.0027) | (0.0027) | (0.0027) | (0.0027) |
| hhh_fem | 0.0137 | 0.2092*** | 0.0524 | 0.0568 | -0.0794 | 0.1070 | 0.0439 |
| | (0.0725) | (0.0711) | (0.0746) | (0.0702) | (0.0720) | (0.0705) | (0.0715) |
| treatment:hhh_fem | 0.2028** | -0.1137 | -0.0660 | -0.0578 | -0.1581 | -0.2036** | -0.0179 |
| | (0.1016) | (0.0994) | (0.1027) | (0.0984) | (0.1006) | (0.0988) | (0.0999) |
| pben_fem | -0.2608 | -0.2878 | -0.0207 | -0.6038*** | -0.0318 | -0.3847** | 0.0823 |
| | (0.1926) | (0.1921) | (0.1927) | (0.1865) | (0.1917) | (0.1872) | (0.1903) |
| treatment:pben_fem | 0.0380 | 0.5083* | 0.4399 | 0.2023 | 0.0754 | 0.7009** | 0.2081 |
| | (0.2927) | (0.2890) | (0.2935) | (0.2835) | (0.2914) | (0.2846) | (0.2892) |
| hou_hea_min | -0.0004 | 0.0004 | 0.0008 | 0.0002 | -0.0011* | 0.0000 | 0.0002 |
| | (0.0006) | (0.0006) | (0.0006) | (0.0006) | (0.0006) | (0.0006) | (0.0006) |
| treatment:hou_hea_min | -0.0006 | -0.0009 | -0.0014* | -0.0023*** | 0.0006 | -0.0006 | -0.0009 |
| | (0.0008) | (0.0008) | (0.0008) | (0.0008) | (0.0008) | (0.0008) | (0.0008) |
| hou_mar_min | 0.0002 | -0.0007* | -0.0005 | -0.0005 | 0.0003 | -0.0004 | -0.0001 |
| | (0.0004) | (0.0004) | (0.0004) | (0.0004) | (0.0004) | (0.0004) | (0.0004) |
| treatment:hou_mar_min | 0.0003 | -0.0002 | 0.0007 | 0.0003 | -0.0003 | 0.0008 | -0.0011* |
| | (0.0006) | (0.0006) | (0.0006) | (0.0006) | (0.0006) | (0.0006) | (0.0006) |
| hou_wat_min | -0.0010 | 0.0000 | -0.0011 | -0.0011 | 0.0017 | 0.0023** | -0.0010 |
| | (0.0011) | (0.0011) | (0.0011) | (0.0011) | (0.0011) | (0.0011) | (0.0011) |
| treatment:hou_wat_min | -0.0002 | 0.0016 | 0.0009 | -0.0017 | -0.0052*** | -0.0039** | 0.0000 |
| | (0.0017) | (0.0017) | (0.0017) | (0.0016) | (0.0017) | (0.0017) | (0.0017) |
| R-squared | 0.0099 | 0.0174 | 0.0340 | 0.0366 | 0.0120 | 0.0324 | 0.0150 |
| R-squared Adj. | 0.0065 | 0.0140 | 0.0305 | 0.0333 | 0.0086 | 0.0291 | 0.0117 |

Table 6: Out-of-sample evaluation of optimized policy value (**higher** is better). Rows indicate policies optimizing different estimates on the training set. Columns indicate different evaluation methods on the test set. Estimated factors $\hat{Z} = 2$. We report the mean $\pm$ standard deviation. Bold is best (within-column), italic second-best.

| $\pi^*$ | Self | $\rho^\top \mu^{RR-DM}$ | $\rho^\top Y^{IPW}$ | $\rho^\top Y^{DR}$ |
|---|---|---|---|---|
| $\rho^\top \mu^{DM}$ | $0.101 \pm 1.836$ | $-0.126 \pm 1.037$ | $\mathbf{0.180} \pm 11.959$ | $\mathbf{0.177} \pm 11.908$ |
| $\rho^\top \mu^{RR-DM}$ | $-0.006 \pm 0.909$ | $\mathbf{-0.006} \pm 0.909$ | $\mathit{0.144} \pm 11.461$ | $\mathit{0.147} \pm 11.407$ |
| $\rho^\top Y^{IPW}$ | $-0.224 \pm 10.120$ | $-0.330 \pm 1.026$ | $-0.224 \pm 10.120$ | $-0.225 \pm 10.094$ |
| $\rho^\top \mu^{RR-IPW}$ | $-0.045 \pm 1.227$ | $\mathit{-0.044} \pm 0.919$ | $0.046 \pm 11.217$ | $0.046 \pm 11.166$ |
| $\rho^\top Y^{DR}$ | $-0.258 \pm 10.244$ | $-0.305 \pm 1.029$ | $-0.253 \pm 10.276$ | $-0.258 \pm 10.244$ |
| $\rho^\top \mu^{RR-CV}$ | $-0.045 \pm 1.231$ | $\mathit{-0.044} \pm 0.919$ | $0.046 \pm 11.217$ | $0.046 \pm 11.166$ |

Table 7: Out-of-sample evaluation of optimized policy value (**higher** is better). Rows indicate policies optimizing different estimates on the training set. Columns indicate different evaluation methods on the test set. Estimated factors $\hat{Z} = 3$. We report the mean $\pm$ standard deviation. Bold is best (within-column), italic second-best.

| $\pi^*$ | Self | $\rho^\top \mu^{RR-DM}$ | $\rho^\top Y^{IPW}$ | $\rho^\top Y^{DR}$ |
|---|---|---|---|---|
| $\rho^\top \mu^{DM}$ | $0.101 \pm 1.836$ | $0.083 \pm 1.548$ | $0.180 \pm 11.959$ | $0.149 \pm 11.851$ |
| $\rho^\top \mu^{RR-DM}$ | $0.168 \pm 1.609$ | $\mathbf{0.168} \pm 1.609$ | $\mathbf{0.363} \pm 12.557$ | $\mathbf{0.315} \pm 12.437$ |
| $\rho^\top Y^{IPW}$ | $-0.224 \pm 10.120$ | $-0.255 \pm 1.672$ | $-0.224 \pm 10.120$ | $-0.199 \pm 10.098$ |
| $\rho^\top \mu^{RR-IPW}$ | $0.195 \pm 2.376$ | $\mathit{0.151} \pm 1.639$ | $\mathit{0.321} \pm 12.348$ | $\mathit{0.278} \pm 12.240$ |
| $\rho^\top Y^{DR}$ | $0.008 \pm 10.275$ | $-0.208 \pm 1.682$ | $-0.025 \pm 10.355$ | $0.008 \pm 10.275$ |
| $\rho^\top \mu^{RR-CV}$ | $0.194 \pm 2.381$ | $\mathit{0.151} \pm 1.639$ | $\mathit{0.321} \pm 12.349$ | $\mathit{0.278} \pm 12.241$ |

Table 8: Out-of-sample evaluation of optimized policy value (**higher** is better). Rows indicate policies optimizing different estimates on the training set. Columns indicate different evaluation methods on the test set. Estimated factors $\hat{Z} = 5$. We report the mean $\pm$ standard deviation. Bold is best (within-column), italic second-best.

| $\pi^*$ | Self | $\rho^\top \mu^{RR-DM}$ | $\rho^\top Y^{IPW}$ | $\rho^\top Y^{DR}$ |
|---|---|---|---|---|
| $\rho^\top \mu^{DM}$ | $0.101 \pm 1.836$ | $\mathbf{0.128} \pm 1.723$ | $\mathbf{0.180} \pm 11.959$ | $\mathbf{0.160} \pm 11.772$ |
| $\rho^\top \mu^{RR-DM}$ | $0.117 \pm 1.765$ | $\mathit{0.117} \pm 1.765$ | $\mathit{0.149} \pm 11.928$ | $0.125 \pm 11.719$ |
| $\rho^\top Y^{IPW}$ | $-0.224 \pm 10.120$ | $-0.326 \pm 1.822$ | $-0.224 \pm 10.120$ | $-0.217 \pm 10.103$ |
| $\rho^\top \mu^{RR-IPW}$ | $0.081 \pm 2.498$ | $0.079 \pm 1.733$ | $0.129 \pm 11.632$ | $\mathit{0.127} \pm 11.458$ |
| $\rho^\top Y^{DR}$ | $0.050 \pm 10.445$ | $-0.278 \pm 1.814$ | $0.028 \pm 10.546$ | $0.050 \pm 10.445$ |
| $\rho^\top \mu^{RR-CV}$ | $0.082 \pm 2.505$ | $0.079 \pm 1.733$ | $0.130 \pm 11.632$ | $\mathit{0.127} \pm 11.459$ |

Table 9: Out-of-sample evaluation of optimized policy value (**higher** is better). Rows indicate policies optimizing different estimates on the training set. Columns indicate different evaluation methods on the test set. Estimated factors $\hat{Z} = 6$. We report the mean $\pm$ standard deviation. Bold is best (within-column), italic second-best.

| $\pi^*$ | Self | $\rho^\top \mu^{RR-DM}$ | $\rho^\top Y^{IPW}$ | $\rho^\top Y^{DR}$ |
|---|---|---|---|---|
| $\rho^\top \mu^{DM}$ | $0.101 \pm 1.836$ | $\mathbf{0.090} \pm 1.774$ | $\mathbf{0.180} \pm 11.959$ | $\mathbf{0.151} \pm 11.767$ |
| $\rho^\top \mu^{RR-DM}$ | $0.081 \pm 1.777$ | $\mathit{0.081} \pm 1.777$ | $\mathit{0.168} \pm 11.919$ | $\mathit{0.137} \pm 11.733$ |
| $\rho^\top Y^{IPW}$ | $-0.224 \pm 10.120$ | $-0.285 \pm 1.835$ | $-0.224 \pm 10.120$ | $-0.212 \pm 10.119$ |
| $\rho^\top \mu^{RR-IPW}$ | $0.068 \pm 2.432$ | $0.047 \pm 1.750$ | $0.124 \pm 11.859$ | $0.103 \pm 11.723$ |
| $\rho^\top Y^{DR}$ | $-0.038 \pm 10.165$ | $-0.239 \pm 1.851$ | $-0.069 \pm 10.240$ | $-0.038 \pm 10.165$ |
| $\rho^\top \mu^{RR-CV}$ | $0.067 \pm 2.442$ | $0.047 \pm 1.750$ | $0.124 \pm 11.860$ | $0.103 \pm 11.723$ |

