# OpenReview forum: "Reduced-Rank Outcome Compression for Causal Policy Optimization"
_TMLR — Accepted by TMLR_

### Review · Reviewer_ojc5 · 2026-01-18

**Summary Of Contributions:**

This paper addresses the challenge of learning optimal treatment policies when researchers collect multiple noisy outcomes that are imperfect measurements of the underlying true object of interest. The authors propose using reduced-rank regression (RRR) to denoise high-dimensional outcomes by recovering low-rank structure, then develop variance-reduced estimators (direct method, IPW, and control variate variants) for policy evaluation and optimization. They establish unbiasedness and consistency results for their proposed estimators, along with finite-sample generalization bounds. They demonstrate improvements in both simulated experiments and a real-world cash transfer RCT dataset from the Sahel region.

# Strengths:
$1.$ The paper tackles a genuinely important and underexplored issue in causal policy learning. Multiple noisy outcomes are common in social science applications, and the connection to commonly-used index variables in development economics is compelling.

$2.$ The use of reduced-rank regression as a structured dimensionality reduction technique is well-suited to the problem.

$3.$ The paper makes solid theoretical contributions and is supported by a thorough empirical evaluation.

# Weaknesses:

$1.$ Strong linearity assumptions. While the robustness checks in Section 4.1 are appreciated, the core methodology assumes linear relationships between covariates, latent factors, and outcomes, which is a strong assumption. Future work could address the consequences of the violation of this assumption and further check the robustness of the methodology in more experiments.

$2.$ Scalarization weights ρ are assumed given. The paper sidesteps a fundamental challenge: how do policymakers choose ρ? This is acknowledged but not adequately addressed.

**Additional Comments:**

NA

**Audience:**

Yes

**Audience Explanation:**

This paper addresses a problem at the intersection of causal ML, machine learning, and econometrics. The methodology would be valuable for applied researchers seeking to improve their estimates in real-world settings (e.g. development economics and healthcare) where multiple noisy outcomes are common.

**Claims And Evidence:**

Yes

**Claims Explanation:**

The core claims regarding variance reduction from reduced-rank regression are well-supported by consistent evidence across experiments, with clear visualizations. The theoretical properties are rigorously established with mathematical proofs.

**Requested Changes:**

$1.$ When introducing $\rho$ at the end of page 2, it is a bit out of context. One sentence explaining the goal of this term (like at the end of page 4) would help the readers understand the intuition.

$2.$ As currently written, Assumption 1 is trivially satisfied for any matrix $C$, since $rank(C)≤min⁡(k,p)$ always holds. It may be helpful to rewrite the assumption to explicitly emphasize a low-rank condition (e.g., $rank(C) << min(k,p)$ ) , or clarify whether this is the intended interpretation.

$3.$ Assumption 3 should explicitly specify the distributional assumptions or moment conditions on the noise terms $U$ and $\varepsilon$.

$4.$ Before lemma 6, it is stated that the estimator is asymptotically unbiased, while the lemma states that it is unbiased. Could you clarify the ambiguity?

$5.$ In Algorithm 1, the updates should take place on $\theta$, i.e. the parameters of the policy $\pi_{\theta}$, not the policy itself.

$6.$  The statement “Control variate methods address this by adding zero-mean terms that are correlated (or anti-correlated) with the random quantities being averaged, thereby reducing variance without introducing bias. “ could benefit with a citation.

$7.$ There is a missing dot in “rank r ) Empirically” in page 7.

$8.$ Theorem 13 would benefit from having assumptions 17 and 18 stated immediately prior to it.

$9.$ In page 8 and in page 14, could you please clarify how or why $\rho$ was chosen to be $[0.3987, 0.0212, \dots]$ and  $[8.82, 1.31, \dots]$ ?

$10.$ in Figure 1, the right plot is missing half of the lines.

---

> ### Author Response · Authors · 2026-02-19
>
> Thank you for your very constructive feedback. We appreciate that you have recognized the ubiquity and importance of the problem. We respond point-by-point below and describe concrete revisions we have made.
>
> Since TMLR recommends uploading only one revision after all reviews are in, we will hold off on uploading the updated pdf until then.
>
> ## Weaknesses
>
> >  1. Strong linearity assumptions...
>
> We would like to clarify both the motivation for these assumptions and the robustness of our method to their violation. First, these modeling choices are motivated by the nature of the application domains we study (e.g., social and administrative data), where outcomes are high-dimensional, noisy, and often driven by a small number of latent factors.
>
> There is also theoretical precedent supporting the fact that these assumptions can apply beyond linear settings. For example, Lin et al. [1] show that ordinary least squares with full treatment–covariate interactions is consistent for ATE estimation.
>
> **Empirical robustness to linearity:** As you mention, our empirical results and robustness checks demonstrate that our estimator remains robust under nonlinearity, such as when the latent factors and outcomes are generated through nonlinear transformations. We illustrate the robustness to nonlinearity in our experimental results displayed in Figure 4.
>
> **Extension to nonlinear factor models:** Our method also seamlessly allows for the low-rank linear outcome model assumption to be replaced with a nonlinear low-rank alternative without altering the IPW structure of the estimator. If a modeler finds that the linearity assumption no longer holds for their data, than one might look to other nonlinear models such nonlinear PCA or nonlinear Matrix Completion to denoise the outcome variables. All this would require is either replacing the plug-in estimators (see eq. 6 and eq. 7 on page 4) derived from the linear low rank model assumptions with nonlinear low rank model assumptions. The other option is to replace how we learn the control variate in Def. 7 on page 6. We have added a paragraph clarifying this extension and sketching out how one might incorporate a nonlinear outcome model in the outcome regression component while preserving the benefits of inverse propensity weighting and variance reduction.
>
> **Violations of linearity:** Using inverse probability weighting (IPW) plays a critical role in mitigating the consequences of model misspecification by correcting for selection bias and substantially reducing the variance of our estimators. For example, it is possible to test for violations of the linearity assumption via standard tests for model misspecification (F-test, etc). If the linearity assumption is violated, then we do not recommend denoising the original outcomes $Y$. Even when the outcome model is imperfectly specified, the DR estimator with the original outcomes $Y$ still provides bias-correction and a greater reduction in the variance.
>
> Finally, trying to nonparametrically learn multiple outcome without imposing some relationship between the variables would just make the already existent variance challenges in nonparametric outcome modeling much worse. We mention nonlinear extensions as a promising direction for future work, but we highlight that there is still a lot to be gained by implementing our proposed variance reduction estimator.
>
> We propose the following paragraphs in the conclusion that can discuss these points on linearity violations and extensions to nonlinearity:
>
> > We examined the statistical challenges that arise when learning policies with multiple outcomes and proposed a variance-reduction framework based on low-rank structure in multivariate regression. By combining dimensionality reduction with inverse propensity weighting (IPW), our estimator improves policy evaluation and optimization while preserving robustness to treatment assignment mechanisms. Importantly, our framework does not rely fundamentally on linearity. The low-rank linear outcome model can be replaced with a nonlinear low-rank alternative without altering the IPW structure of the estimator. If linearity is violated, practitioners may instead employ nonlinear PCA, nonlinear matrix completion, or other nonlinear low-rank models to denoise the outcome vector. Operationally, this requires only replacing the plug-in estimators derived under the linear low-rank assumption (Eqs. 6–7) or modifying how the control variate in Def. 7 is learned.
> > Moreover, when linearity fails, we recommend a conservative approach: use the doubly robust (DR) estimator with the original outcomes $Y$ rather than aggressively denoising via a misspecified model. Standard diagnostics (e.g., F-tests for model misspecification) can assess linearity violations. If linearity is violated, then even under imperfect outcome modeling, the DR estimator can still provide bias correction and variance reduction benefits.
>
> [1] Lin, W. (2013). Ann. Appl. Stat.

---

> ### Author Response · Authors · 2026-02-19
> **Author Responses Cont'd**
>
> >  Scalarization weights...
>
> Yes, this is important. This remains an open question for multi-objective optimization in general. We propose one remedy to highlight how existing common practices of _index variables_ can be mapped onto a $\rho$ vector. Given this mapping of $\rho$ to existing common practices, policy-makers may be able to reason about variations thereof. Our specification of the preference vector in the model emulates these common practices used in economic literature.
>
> We propose to add the following paragraph to the revision, which walks through existing policymaker choices of an index variable that weights multiple important outcomes of interest, e.g. our $\rho^\top Y$, drawing on the broader context of the Sahel dataset.
>
> > Let's walk through two example constructions of an index variable in this setting. In poverty alleviation programs, which is our main motivation and data source, policy makers are tasked with trying to determine if their interventions are improving outcomes. In the study conducted by Bossuroy et al [4], they measure a multitide of outcomes and combine them by specifying a $\rho$ vector and summing across the outcomes. $\rho$ can often be a simple vector of 1's where each outcome is weighted equally and summed. For example, in the Sahel dataset, there is an index called 'Self efficacy' defined as the sum of answers to eight specific survey questions. Another example of an index called 'Dietary diversity', where the index is defined as a linear combination of the number of days that individuals consumed a produce. The index looks as follows: 2 $\times$ cereals + 2$\times$tubers + 3$\times$pulses + vegetables + fruit + 4$\times$meat/fish/eggs + 4$\times$milk + 0.5$\times$oil + 0.5$\times$sugar. Both of these indices have been well studied in the psychology [6] and nutrition [5] literature. They illustrate that policy makers are currectly specifying some version of $\rho$ whether implicitly or explicitly. Other related works such as Luckett et al. [2] and Lizotte et al. [3] both highlight methods that use expert-derived combined outcomes but say that these methods do not account for differences across units. Ultimately, the methodology that we are proposing in this paper does not depend on the choice of $\rho$. Our goal is to denoise outcomes in order to improve causal estimation and policy learning. Our experiements in Figure 2 show that this is achieved.
>
> We included these concrete examples of the index variables to help make clear the purpose of $\rho$ in our revision.
>
> [2] Luckett, D. et al. (2021). JMLR
>
> [3] Lizotte, D. et al. (2012). JMLR
>
> [4] Bossuroy, T., et al (2022). Nature
>
> [5] WFP. Food Consumption Analysis: Calculation and Use of the Food Consumption Score in Food Security
> Analysis 2008.
>
> [6] Radloff, L. S. The Ces-D Scale: A Self-Report Depression Scale for Research in the General Population.
> Applied psychological measurement 1, 385–401 (1977).
>
> ## Responses to requested changes 1-4
>
> 1. When introducing $\rho$ at the end of page 2, it is a bit out of context. One sentence explaining the goal of this term (like at the end of page 4) would help the readers understand the intuition.
>
> Thank you for this suggestion. We do go into more detail in Section 2.3 about $\rho$, but we recognize that this explaination and context is a bit late in the paper.
>
> To fix this, we will add a couple of sentences early on to help the readers understand the intuition and goal. We will add the following sentence:
>
> > In both settings, a weighting vector $\rho$ encodes decision-maker preferences by specifying how multiple observed outcomes or latent factors are aggregated into a single scalar objective for policy learning. The goal of this work is to denoise the outcomes separate from the scalarization.
>
> 2. As currently written, Assumption 1 is trivially satisfied for any matrix $C$, since $rank(X) \leq min(k,p)$ always holds. It may be helpful to rewrite the assumption to explicitly emphasize a low-rank condition (e.g., $rank << min(k,p)$) , or clarify whether this is the intended interpretation.
>
> We thank the reviewer for pointing this out, we will be sure to update our low-rank condition as follows:
>
> > \textbf{Assumption 1} (Low rank regression coefficients). $rank(C) = r << min(k, p)$.
>
> 3. Assumption 3 should explicitly specify the distributional assumptions or moment conditions on the noise terms $U$ and $\epsilon$.
>
> We thank the review for bringing this to our attention. We assume that each of the noise terms are mean-zero normal distributions. We will be sure to update the manuscript.
>
>
> 4. Before lemma 6, it is stated that the estimator is asymptotically unbiased, while the lemma states that it is unbiased. Could you clarify the ambiguity?
>
> We mean to report that this estimator is both unbiased in finite-sample guarantees and asymptotically unbiased as the sample size grows. We will remove the text above Lemma 6 so as to not confuse the reader.
>
> [more in next comment]

---

> > ### Author Response · Authors · 2026-02-19
> > **Author Responses Cont'd x2**
> >
> > ## Remaining responses to requested changes 5-10
> >
> > 5. In Algorithm 1, the updates should take place on  $\theta$, i.e. the parameters of the policy $\pi_{\theta}$
> > , not the policy itself.
> >
> > Thanks for catching this. It is a small adjustment that we have made in the revision, which will be uploaded after all reviews are in.
> >
> >
> > 6. The statement “Control variate methods address this by adding zero-mean terms that are correlated (or anti-correlated) with the random quantities being averaged, thereby reducing variance without introducing bias. “ could benefit with a citation.
> >
> > Thanks for this helpful comment. We will be sure to add our main citation for control variates to this sentence specifically:
> >
> > Peter W. Glynn and Robert Szechtman. Some new perspectives on the method of control variates. Departmentof Management Science and Engineering, Stanford University, Stanford CA 94305, USA, 2002.
> >
> > 7. There is a missing dot in “rank r ) Empirically” on page 7.
> >
> > Thank you for the catching this minor error. We will be sure to add the period to the final manuscript.
> >
> > 8. Theorem 13 would benefit from having assumptions 17 and 18 stated immediately prior to it.
> >
> > Thank you for your helpful comments. To make sure that the manuscript is clear, we will move Assumptions 17 and 18 to just above Theorem 13.
> >
> > 9. In page 8 and in page 14, could you please clarify how or why $\rho$ was chosen to be?
> >
> > In our experiments, $\rho$ is randomly drawn from a normal distributions (standard normal and truncated normal with higher variance). We choose $\rho$ randomly to showcase the utility of denoising regardless of the choice of $\rho$. In practice, experts will specify this weighting vector to determine which outcomes are more or less important to the target construct. Another common choice of $\rho$ is to weight each outcome equally. The focus of this work is not on the choice of $\rho$. In some sense, it does not really make a difference for our proposed methodology.
> >
> > 10. In Figure 1, the right plot is missing half of the lines.
> >
> > The lines in this plot are not missing, they are just behind the other lines. All of the different variants of the CV estimator that we propose perform similarly as the sample size increases. We will make that explicit in the caption for the figure.

---

### Review · Reviewer_fVzp · 2026-02-05

**Summary Of Contributions:**

This paper proposes an estimator for policy evaluation (and hence, a means of policy optimization) in the setting of multiple outcomes. This paper focuses on the setting with a set of outcomes (so $Y$ is multi-dimensional), but domain experts have already specified a weighting vector $\rho$ such that the outcome to optimize is $\rho^\top Y$.  The core proposal is two-fold:  The first component is to use reduced rank regression, on the assumption that potential outcomes are generated as $Y(t) = A_t Z(t) + \epsilon$, where $Z(t)$ is a lower-dimensional set of latent outcomes that itself is a linear function of $X$ via $Z(t) = B_t X + U$.  With this approach in mind, the second component of the proposal is a control variate estimator that learns an optimal weighting of control variates.  The estimators are all shown to be consistent.  The primary comparison between estimators is done empirically on simulated data and data from the "Sahel" dataset, evaluating a randomized trial evaluating social safety net programs.

**Additional Comments:**

## Minor comments on the motivation

One minor point which I found confusing on a first read: In the introduction, it sounded like one of the contributions of the paper will be to help reduce the need for choosing arbitrary scalarization weights (see "Third, having many outcomes can reduce actionability by requiring decision makers to specify or choose scalarization weights"), but a core assumption (Assumption 4) is that such weights are already chosen.  It's a little unclear to me what point is trying to be made here as a result.

## Minor comments on the theoretical presentation

It was not clear to me if the propensity score is assumed to be known:  If so, it would be nice to state this assumption explicitly.  All estimators that use the propensity score use the notation $e_t$ (without a hat), and Algorithm 1 (which uses the IPW estimator defined in Lemma 6) only mentions estimation of an outcome model, not a propensity score.

The use of $\mathbb{E}$ (the population expectation) and $\mathbb{E}_n$ (the empirical expectation) is inconsistent in places, leading to confusion.  For instance, in Section 3, the DM and IPW estimators are introduced as empirical estimators (using $\mathbb{E}_n$), but Lemmas 5 and 6 use $\mathbb{E}$ in the inner expectation.

Another minor point of confusion in Section 3.1:  $D_t^\star$ is defined with respect to $C_t$, but $C_t$ itself is defined with respect to $h_t(X) = \hat{B}_t X$, and so in some sense $D_t^\star$ is itself a "moving target", changing as $\hat{B}_t$ changes with $n$.  It was unclear to me how this is handled;  E.g., should $D_t^\star$ be defined with respect to the true $B_t$?


## Minor comments on the empirical results

Figure 1 is confusing for several reasons:
* The legend in Figure 1 (right) refers to 6 estimators, but I can only make out three estimators.  Moreover, the legend includes "CV1", "CV2", "CV3" which are not defined anywhere I could see.
* It is very difficult to compare across the two plots, given the extreme difference in the $X$-axis, which seems to mainly be an artifact of the IPW estimator using $Y$ having extremely high variance.
* Figure 1 is described in the text as "compar[ing] the variance of all the estimators for the ATE", but it seems to actually show the variance of the pseudo-outcome?  Otherwise, it is very confusing why the "variance" of the IPW estimator does not decrease as the sample size increases.

It is hard to tell how much to read into the results in Figure 2 without some sense of variance in the lines themselves.  Adding error bars to the visualization would be helpful, and could be made small by e.g., running 1000 rather than 100 simulations.

Is there a typo in Table 1, or is it true that $\rho^\top \mu^{DM}$ and $\rho^\top \mu^{RR-DM}$ have exactly the same out-of-sample variance of 4.373?

In the real-data experiment, I found it unclear how to read Table 2;  It would help to move some of the explanation in the paragraph in 4.2.2 up to the table caption.

## Very minor comments

The setup in Section 2 refers to a setting where $Y(t)$ is a deterministic function $Y(t) = g(Z(t))$ of $Z(t)$, but the core Assumption 3 includes noise.

I found the duplicate use of $C$ confusing, as it is used in two places to mean two different things (control variates in Section 3.1, and the RRR model coefficients in Section 2.2).

Theorem 13 refers to Assumptions 17 and 18 in the appendix, which are restatements of the assumptions 11 and 12 above Theorem 13.  This is likely due to a duplicate `\label` in latex.  A similar problem appears where Table 2 is referred to as Table 8?

In Figure 1, the estimator $\rho^\top A Z$ is not defined, though it is easy enough to infer that it refers to (I believe) the "oracle" estimator that knows $A, Z$.

**Audience:**

Yes

**Audience Explanation:**

As instructed by TMLR guidelines, I would err on the side of "perhaps some individuals would be interested in knowing these findings", modulo my concerns regarding evidence above.

**Claims And Evidence:**

No

**Claims Explanation:**

The central claim of the submission is that their suite of estimators improve estimation error in policy evaluation and optimization.  I did not find this claim to be supported by accurate, convincing, and clear evidence.

I have two broad categories of concern, which I describe in more detail under "Requested Changes" and "Additional Comments".  The first concern is that the evidence for improvement over existing approaches is somewhat unclear, perhaps due to a lack of clarity in presentation.  Since the theoretical results don't speak to improvement over other methods (as I understood them), the primary evidence presented is empirical, and a simple baseline is missing in my view.  My second, and perhaps more substantial cross-cutting concern, is a lack of precision and clarity throughout the current submission, in presentation of both theoretical and empirical claims.

**Requested Changes:**

I will use this section to outline my primary concerns, and use the "additional comments" section to lay out a series of places where I found the presentation to be unclear.  Taken in the aggregate, my impression of the paper is that it needs substantial polishing.

## (1) Clearer motivation for the proposed approach beyond RRR for outcome regression

I would request more clarity around why we should expect the proposed estimator to out-perform existing methods (DM, IPW, or DR), since the DR method can itself be viewed as a control-variate approach.

It is certainly the case that, if outcomes are generated according to a low-rank set of coefficients, then we should expect improvement by incorporating that constraint into outcome modeling.

However, once we move beyond "Direct method with reduced rank regression", I find the motivation for other refinements to be confusing. I was quite confused as to the role of the "denoised inverse propensity weighting" estimator in Section 3, which replaces $Y$ in an IPW estimator with $\hat{Y} = \mu_t(X)$.  If the doubly robust estimator tries to get the "best of both worlds", this estimator seems to combine the worst characteristics of the DM and IPW estimator:  It only uses observations $X$ where the on-policy action is taken, can suffer from variance inflation due to small weights, but also depends on a correctly-specified outcome model.

Hence, it is unclear to me why the "denoised IPW" estimator would ever be preferred over the direct method, and why this estimator is the "starting point" for further variance reduction via control variates.  More broadly, there does not appear to be any claim that the control variate estimator with optimal weights is asymptotically better than the doubly-robust method, which contributes to this request for more clarity.

## (2) Comparison to a simple baseline of "dimension reduction"

I would suggest comparing to standard methods (DM, IPW, DR) using the outcome $\tilde{Y} := \rho^\top Y$.  My understanding is that the current instantiation of the "direct method" is fitting separate OLS regressions for each dimension of $Y$, and then taking a linear combination of those predictions.

The rough intuition behind why I would request such a comparison:  The main motivation of the low-rank structure seems to be dimension reduction, but we already have a user-specified means of dimensionality reduction (via Assumption 4), collapsing into a single scalar outcome.

## (3) Revision of theoretical presentation in Section 3.1

I found several areas in the theoretical presentation to be unclear (more on this under "additional comments").  The most salient is that the theoretical presentation in Section 3.1 seems either underspecified or incorrect based on how notation is defined.  Theorem 10 claims that $\sqrt{n} (\phi(\hat{D}\_t) - \phi(D^{\star}\_t)) \rightarrow 0$.  But as I understand from Equation (1), $\phi(D_t)$ is a sample-level observation, and could perhaps more clearly be written as $\phi(X, Y, T; D)$, where $D$ is a learned parameter.  As a result, this statement needs more qualification (e.g., does this convergence to zero happen almost surely over $X, Y, T$?).  I suspect the intention was to make the argument that nuisance function estimation leads to asymptotically negligible error (i.e., that the **estimator** $\hat{V}(\hat{D})$ converges quickly to $\hat{V}(D^*)$), which would be a more reasonable statement.

---

> ### Author Response · Authors · 2026-03-25
>
> We thank the reviewer for their detailed feedback and critical engagement.
>
> We first would like to clarify that in our rebuttal, we have more explicitly stated our central claim (ie in the last paragraph of intro): that denoising and variance reduction are shrinkage-type approaches that can reduce variance in off-policy evaluation and learning. We don't claim universal improvement, although we conduct extensive empirical evaluations to transparently assess the ablations.
>
> Regarding your two broad categories of concern:
> - *Unclear evidence for improvement:* We have clarified that we don't intend _uniform_ improvement from our methods, but rather that denoising and variance reduction can contribute to key *bias-variance* tradeoffs. Also, our submitted draft had already included explicit comparison to standard doubly-robust estimation, so we believe this was a misunderstanding. We demonstrated improvement in finite samples and robustness checks indicate that low-rank assumptions can be helpful despite potential misspecification. Please let us know if you meant something else.
>
> Our new Corollary 5 explicitly highlights this comparison between model bias from denoising, and irreducible outcome variance. To concretize why our motivating applications offer larger opportunities for variance reduction, we have added empirical evidence to the introduction as to the significant residual variance in the Sahel real-world dataset (which is indicative of other real-world data).
>
> - *Lack of precision and clarity:* We appreciate you raising this concern. We have included significant revisions in the Introduction an dMethodology section to clarify our central claims and how our methodological developments relate to them. For example, we have suppressed the discussion of denoised direct method and IPW as natural variants derived "along the way" to our preferred estimator, hence moving some discussion to the appendix - we understand from your review that this caused some confusion. Thank you for pointing out potential notational small fixes to make here -- we have fixed all of these in the revision and undergone polishing throughout. (Not all of the polishing has been offset in blue.)
>
> ## Requested Changes
>
> ### (1) Clearer motivation for the proposed approach beyond RRR for outcome regression
>
> We would like to clarify that the final estimators that we propose are the denoised CV estimators, not the denoised IPW estimator. This final CV estimator has the model-robustness properties that doubly-robust estimation offers.  Our experiments also indicate that denoising, independently, achieves variance reduction across all the estimators. But the CV estimator can still be a theoretically sound practical choice because it offers robustness properties as well.
>
> When dealing with high variance social outcomes, the trade-off between bias and variance becomes very pronounced. Our proposed denoised approaches reduce the variance from irreducible outcome variance, at the cost of introducing some additional bias. Such bias-variance trade-offs are typical of shrinkage approaches that have also been investigated for off-policy evaluation. Our denoising approach targets a non-traditional source of variance from outcome noise. In the revision, we include an extensive discussion about this motivation and provide a corollary to formalize this intuition about bias-variance tradeoffs. We compare conditions as to when denoised IPW indeed could outperform its outcome-dependent counterpart: exactly when the noise from outcome variance is much grader than potential bias introduced from denoising (and some correction terms). (Analogous comparisons for a doubly-robust estimator rely on comparing bias and analogous residual variance).
>
> > More broadly, there does not appear to be any claim that the control variate estimator with optimal weights is asymptotically better than the doubly-robust method, which contributes to this request for more clarity.
>
> The doubly robust estimator is in the class of potential control variates that the regression control variate implicitly optimizes variance over. From standard efficiency theory we know that doubly-robust estimation/AIPW is optimal. RR-DR is feasible inside the regression CV class; meanwhile, RR-CV picks the asymptotic variance-minimizing coefficient inside that class.
>
> Therefore we rather expect that RR-CV can weakly improve upon RR-DR but should rather be thought of as a generalization that is compatible with further denoising, rather than asymptotic improvement, which we indeed do not try to claim. We have added a clarification of this right before Lemma 7.

---

> > ### Author Response · Authors · 2026-03-25
> > **Author Responses Cont'd**
> >
> > ### (2) Comparison to a simple baseline of "dimension reduction"
> >
> > Actually, we'd like to first emphasize that in our experiments, we **already did compare** to $\tilde{Y} := \rho^\top Y$ for the IPW and DR variants. Overall, we find a benefit to using denoised outcomes over the outcomes $Y$.
> >
> > Our approach provides a data-driven approach to variance reduction on top of the user-specified means to improve policy learning and optimization. Often the user-specified means of dimension reduction is chosen in a ad-hoc manner (e.g. averaging outcomes deemed similar a priori), so we provided a data-driven approach leveraging shared structure across multiple outcomes via reduced-rank regression, not just naive OLS. We would also like to clarify that Assumption 4 is about scalarizing multiple outcomes, and reflects preference weighting done by policy-makers, but it is not statistically derived. We have given additional context in the introduction describing how current practitioner approaches based on simple averaging do not leverage the underlying statistical correlation structure underlying multiple outcomes.
> >
> > ### (3) Revision of theoretical presentation in Section 3.1
> >
> > In the uploaded revision draft, we have made significant improvements to the methodology section. First, we fix the notation used in Theorem 10. This was a typo that that was carry-over from an old-notation system. We fix the notation and make the argument that the estimator of the value function $\hat V^{RR-C}(\phi(\hat D;\pi))$ achieves asymptotic normality. With these results, we are not claiming almost sure convergence. In our uploaded revision, we have made significant improvements including a sufficient condition for when denoising can improve as well as improved notational consistency.
> >
> > ## Additional Comments:
> > ### Minor comments on the motivation:
> >
> > We have restructured the introduction to make the motivation and contributions of our paper much clearer. The main claims of our paper are two-fold: (1) Given a user-specified scalarization, we see that our denoising estimators (specifically replacing Y with $\hat Z \hat Y$) achieves lower variance. (2) Designed to emulate the DR approach and give the best of both worlds (that is reducing the variance and being more robust than IPW), our proposed CV estimator offers a strong practical option, that has strong theoretical guarentees in causal inference, and can perform better than the Y-DR variant.
> >
> > ### Minor comments on the theoretical presentation:
> > >It was not clear to me if the propensity score is assumed to be known: If so, it would be nice to state this assumption explicitly. All estimators that use the propensity score use the notation  (without a hat), and Algorithm 1 (which uses the IPW estimator defined in Lemma 6) only mentions estimation of an outcome model, not a propensity score.
> >
> > We updated the revision to have the correct notation for estimated propensity score models. We provide an additional corollary in the appendix (Corollary 22) that shows how the generalization bound changes when propensity scores are estimated.
> >
> > >The use of $\mathbb{E}$ (the population expectation) and
> >  $\mathbb{E}_n$ (the empirical expectation) is inconsistent in places, leading to confusion. For instance, in Section 3, the DM and IPW estimators are introduced as empirical estimators (using $\mathbb{E}$), but Lemmas 5 and 6 use $\mathbb{E}$ in the inner expectation.
> >
> > We updated the revision to fix these inconsistencies. Lemma 5 and Lemma 6 should have used the empirical expectation like it is introduced in the definition of the estimator. We also move these Lemmas to the Appendix to make space in the main paper.
> >
> > >Another minor point of confusion in Section 3.1: $D_t^\*$ is defined with respect to $C_t$, but $C_t$ itself is defined with respect to $h_t(X)$, and so in some sense $D_t^\*$ is itself a "moving target", changing as $\hat{B}_t$ changes with $n$. It was unclear to me how this is handled; E.g., should $D^\*_t$ be defined with respect to the true $B_t$?
> >
> > Yes, $D^\*_t$ is defined with respect to the true $B_t$. We make that clearer in the revision, by defining $C^*_t$ in terms of the true $B_t$.
> >
> > ### Minor comments on the empirical results: duplicates, missing definitions and notation
> >
> > Thanks for catching these issues, we address each of these points in our revision.

---

### Review · Reviewer_svCo · 2026-03-24

**Summary Of Contributions:**

This paper looks at policy learning when the decision-maker observes many noisy, correlated outcomes instead of one clean target. The main idea is to use reduced-rank regression to recover a lower-dimensional signal from those outcomes, and then use that representation for policy evaluation and optimization. The paper also proposes a control-variate estimator built on top of this structure to reduce variance further. Empirically, the method is tested on synthetic data and on a real social-program dataset.

The paper’s main strengths are that it tackles a relevant problem, the method is fairly clean and intuitive, and the experiments are reasonably comprehensive.
The main weaknesses are that the approach depends on a low-rank modeling assumption, and the real-data section is necessarily less conclusive than the simulations because true counterfactual policy value is not observed.

**Audience:**

Yes

**Audience Explanation:**

Yes. I think this paper will be of interest to readers working on causal ML, off-policy evaluation, treatment assignment, and policy learning more broadly. The setting of multiple noisy outcomes is practically important, and the paper offers a fairly simple way to exploit structure across outcomes instead of treating them separately.

**Claims And Evidence:**

Yes

**Claims Explanation:**

Overall, yes. The paper provides both theory and experiments in support of its main claims. On the theory side, it gives results for unbiasedness/consistency and motivates why the proposed control-variate estimator should improve variance. On the empirical side, the synthetic experiments are convincing: they show that denoising the outcomes helps policy evaluation and often improves policy optimization as well. The robustness experiments to nonlinearity are also helpful.

My main reservation is with the real-data evidence. It is useful and directionally supportive, but it cannot fully validate true policy quality because the ground-truth counterfactual policy value is unavailable. So I think the central claims are supported, but the strongest evidence comes from the simulated setting.

**Requested Changes:**

The paper should be clearer about what is new relative to existing work on multi-outcome policy learning, low-rank outcome models, and standard variance-reduction methods. The contribution is there, but the novelty would be easier to assess with a sharper comparison to prior work.

Please expand the discussion of assumptions and failure modes. In particular, when should practitioners expect the low-rank structure to hold, and what happens when it does not?

Add a more detailed discussion of rank selection. The real data section gives an argument, but I would like a clearer practical recommendation and some sensitivity analysis.

Clarify how much the results depend on preprocessing choices such as standardization and on the choice of scalarization weights.

---

> ### Author Response · Authors · 2026-03-25
>
> We thank the reviewer for their helpful feedback on our work.
>
> ### Requested Changes:
>
> **1. clear contribution and relation to existing literature**
>
> We add additional discussion to our paper to highlight the novelty of the contributions of our work compared to existing work on multiobjective policy learning, low-rank outcome models, and standard variance-reduction methods. We would like to highlight that our work is the first to integrate reduced-rank outcome denoising into the policy learning pipeline, yielding estimators that can handle noisy, high-dimensional outcome settings and account for the low-rank structure common in real-world policy evaluation.
>
> We added the following paragraph to the introduction:
>
> >Prior work has explored dimensionality reduction techniques (Mckenzie 2005), highlighted specific challenges posed by multiple outcomes (Ludwig 2017, Bjorkegren 2022), and multiobjective policy learning (Boominathan 2020), but exploiting the low-rank structure of outcomes for denoising has not been explored in the policy learning literature$^1$. Relative to prior work on low-rank outcome models (Agarwal 2020) based on model-based estimation alone, we explore rank reduction for causal estimators that incorporate inverse propensity reweighting. Relative to standard variance reduction from double-robustness or shrinkage-based approaches that clip propensity weights to trade-off bias and variance (Su 2020 and Wang), we explore further denoising of factual outcomes to reduce variance from irreducible outcome noise. Our work is the first to integrate reduced-rank outcome denoising directly into the policy learning pipeline, yielding estimators that can handle noisy, high-dimensional outcome settings common in real-world policy evaluation.
>
> > $^1$ Alternative factor models that have been considered, such as probabilistic PCA (Tipping 1999) and penalized factor models (Yuan 2007). We discuss these in detail in Appendix A.1.
>
> **2. assumptions and failure modes**
>
> We add this additional discussion under Section 4.1's simulated experiments and the "Robustness to Nonlinearity" robustness check. The motivating setting that we are targeting with this work is when practitioners are dealing with social data where measurement error is going to be a challenge. It is difficult to define and measure social constructs and so in practice multiple outcomes are collected to represent lower dimensional constructs. When this structural assumption does not hold or our linear assumption fails, we see that our method still performs reasonably well in our robustness checks.
>
> We added the following paragraphs and example instantiation for practitioners to the discussion on robustness to the low rank and linearity assumptions:
>
> >Importantly, when the low-rank assumption fails such as when the true rank is high or the outcome relationships are highly nonlinear, then our method degrades gracerfully and trades off some model bias for variance reduction, as the trade-off analysis of Corollary 5 indicates. In such cases, the variance reduction from denoised outcomes via the reduced-rank model can still yield competitive results in terms of mean squared error. Crucially, the low-rank assumption is well-motivated by the substantive structure of the social data settings we aim to target. In policy evaluation and optimization with many social outcomes, it is common for high-dimensional outcome vectors to be driven by a small number of latent factors, whether due to correlated behavioral responses or common underlying social constructs. Reduced-rank regression is precisely designed to exploit this structure, and our method is tailored to settings where this low-dimensional signal is present but obscured by noise and measurement error.
>
> > To make this more concrete, let us revisit a generic multiple-outcomes setting of a large development randomized-controlled trial, discussed in the introduction and our later real-world data. Low-rank assumptions may true for a portfolio of outcomes based on consumption (as broken down into various categories) savings, income and investments: these are all plausibly explained as linear combinations of a few factors: income, expenses, and overall consumption. Low-rank assumptions are less likely true when a portfolio of outcomes includes outcomes that weakly depend on everything else. For example, suppose multiple outcomes include a comprehensive set of 1-year endline surveyed outcomes spanning well-being/health/employment/education/income and longer-term 5-year income measures measured in administrative data: long-term 5-year income likely depends weakly on all such 1-year endline outcomes, and therefore low-rank is an approximation, not the underlying truth. However, if long-term 5-year income depends much more on a few factors like earlier employment/education/income, as the surrogate index literature argues (Athey et al., 2019), low-rank may be a useful variance-reducing approximation.

---

> > ### Author Response · Authors · 2026-03-25
> > **Author Responses Cont'd**
> >
> > **3. Add a more detailed discussion of rank selection. The real data section gives an argument, but I would like a clearer practical recommendation and some sensitivity analysis.**
> >
> > We add more details in the revision to our discussion of model selection in the experiments section that clarifies the practical recommendation and highlights the results of the sensitivity analysis that we conducted in the Appendix. In summary, we suggest computing the estimated policy values across a range of potential rank values, and selecting the rank that minimizes the estimated policy value among them. Another approach would be to compute the variance explained on a test set and choose the rank value that maximizes this value.
> >
> > We added the following paragraph to the discussion on model selection in Section 4, Experiments:
> >
> > >Based on this analysis, our practical recommendation for rank selection is to select the rank that minimizes the estimated policy value. In our model selection experiements, we also see that this choice is aligned with the rank that maximizes the weighted variance explained on the validation set. The rank achieving the results in Table 3 both maximizes the policy value and explains the most variance in outcomes - suggesting these two ways of rank selection may align well. The rank we chose also achieves the best out-of-sample performance on the evaluation set. However, even if we had chosen a different value of the rank, our results in Table 6-9 show that our method has the second-best performance, suggesting some robustness to the exact choice of rank, and general benefits of variance reduction.
> >
> >
> > **4. Clarify how much the results depend on preprocessing choices such as standardization and on the choice of scalarization weights.**
> >
> > Re: standardization: We suggest standardizing the data and include that as part of our algorithm. Standardization is important and pretty common in this setting and most ML pipelines. Standardizing either the features, outcomes, or both are the only ways to perform this procedure and we do both in our algorithm; so there are not further degrees of freedom there. Standardization ensures that reduced-rank variance reduction doesn't simply target outcomes that happen to be measured on a larger scale.
> >
> > Re: scalarization: our results are not dependent on the choice of scalarization weights. In fact, we choose the weights randomly (of fixed total norm) to help illustrate this. We try to model the real-world scenario where decision-makers chose relative weights based on their experience or prior studies in Assumption 4. However, even with these choice made, the outcomes can still be high variance. Our method is meant to be an improvement upon these expertly-chosen weights in reducing the variance of our value estimator for policy optimization.
> >
> > We added the following sentences in Section 2.3, Multi-objective evaluation and learning to clarify the role of the choice of scalarization weights:
> >
> > > Our methods reduce variance in evaluating and optimizing such policies. This is meant to be an improvement over existing index-based approaches commonly used in economics and multi-objective policy learning. Later, in our experiments, we simulate random values for $\rho$ to model this setting in order to illustrate that our method does not depend on the choice of weighting vector, rather it performs variance reduction on top of the dimension reduction that scalarization provides.
> >
> >
> > and we add the following sentences to Section 4.1, Policy  evaluation to clarify the discussion on standardization:
> >
> >
> > > Without standardization of the features and the outcomes, our methods suffer from the common issues seen with regularized regression and gradient descent optimization where larger values shrink disproportionally compared to smaller values of data.

---

### Author Response · Authors · 2026-03-25
**Response to all reviewers**

We are grateful to reviewers for feedback on our draft. We have uploaded a revision with changes marked in blue with some edits to clarify points of confusion. In addition to minor edits and clarifications in response to reviewer questions, where we have responded directly, we would like to highlight our major revisions:

- We have added more detail in the introduction highlighting the extent of irreducible outcome variance in real-world data, including in the Sahel dataset in the later real-world experiments.
- In section 3, Methodology: We have streamlined our Methodology section. We study two different design changes: denoising, and regression control variates (as a control variate framework compatible with denoising). As a result, introducing these for causal inference results in a number of variants that we ablate later on in the empirics. We have streamlined our presentation and compressed preliminary discussions of denoised direct method and IPW, which we must introduce for a self-contained presentation but are not our core methodological proposal.
- In section 3, Methodology: We have clarified our motivations in introducing denoising: a "shrinkage" method for off-policy evaluation and learning trades off potential model bias in the reduced-rank regression for potential variance gains, in reducing the contribution of outcome variance to MSE. We have introduced a MSE comparison of denoised vs. standard IPW to highlight these key tradeoffs.
- We have added additional discussion of robustness to violations of assumptions, including an example and instantiation for practitioners in the context of the Sahel dataset we discussed in introduction/real-world data.
- We have polished the draft throughout (not all of the small fixes have been highlighted in blue.)

---

### Decision · Action_Editor_ex4Q · 2026-05-11

**Recommendation:** Accept with minor revision

**Additional Comments:**

Please consider addressing the following reviewer comments in your revision:

1. The (new) claim on page 8, under Equation (14), states that
> the standard doubly-robust estimator is within the class of control variates that the regression control variate implicitly optimizes over...therefore, [our method] can weakly improve upon standard DR, for example in fintie [sic] sample, while falling back to asymptotically optimal estimators

The paper defines two estimators using the notation "RR-CV" (in Equation (11) using the original outcomes, and in Equation (14) using the imputed outcomes from the RRR). If this claim applies, it should only apply to the former, not the latter, but the latter estimator in Equation (14) is the main estimator introduced.

2. The newly added Corollary 5 uses notation like $m_t(X)$ that is only defined in the Appendix (for proposition 15). Moreover, the conclusion is difficult to interpret, since it states an improvement when the left-hand-side of the inequality is less than the right-hand-side, but the left-hand-side involves a factor of that the right does not. Digging into the Appendix, it is clear that this is a statement about bias^2 + variance of one method being lower than the variance of another, but that was not very clear (in my view) from presentation in the main paper.

**Audience:**

Yes

**Audience Explanation:**

Yes, the topic is clearly relevant to the TMLR audience.

**Claims And Evidence:**

Yes

**Claims Explanation:**

After revision, 2/3 reviewers agree while one reviewer remains hesitant. The remaining concerns mostly concern presentation, which can be addressed with a minor revision.

---

> ### Author Response · Authors · 2026-06-09
>
> We greatly appreciate the review team's efforts, which have helped us improve the paper.
> We have uploaded a camera-ready that addresses the above reviewer comments. To aid comparison, we also updated a revision with relevant edits highlighted in purple. (Our uploaded camera-ready revision does not have any highlighting).
>
> 1. We have clarified the distinction between the RR-CV-$Y$ estimator in eq 11 and the RR-CV-$\hat Y$ in eq 14 using the imputed outcomes, and we have more clearly stated the claim that RR-CV-$Y$ shares this relationship with standard DR guarantees, while RR-CV-$\hat Y$ does not. These adjustments are on p.8-9 (minor adjustments elsewhere). Overall we have adjusted our language to present both these estimators: RR-CV-$Y$ enjoys stronger theoretical guarantees while our experiments indicate that RR-CV-$\hat Y$ shrinks more aggressively and does well in tailored experiments.
>
> 2. Yes, the statement is a comparison of MSE (bias^2 + variance) of IPW vs. denoised IPW. Although the original text indicated the MSE comparison inline, we have added an additional line indicating the bias^2 + variance decomposition (and therefore making more clear which terms correspond to which). Indeed, the final line separates the prediction bias terms on the left hand side from the outcome noise variance terms on the right hand side, and these are different: it ultimately depends on the data-generating process whether the inequality holds, it is not uniformly valid. However, this explicitly highlights what needs to be established for denoised IPW to improve.